# Values and preferences related to workplace mental health programs and interventions: An international survey

Jill K. Murphy[1‡]*, Jasmine M. Noble[2,3‡], Promit Ananyo Chakraborty[4], Georgia Michlig[5], Erin E. Michalak[1], Andrew J. Greenshaw[2], Raymond W. Lam[1]

1 Faculty of Medicine, Department of Psychiatry, University of British Columbia, Vancouver, Canada,
2 Faculty of Medicine and Dentistry, Department of Psychiatry, University of Alberta, Edmonton, Canada,
3 Faculty of Science, Department of Computing Science, University of Alberta, Edmonton, Canada,
4 Faculty of Medicine, School of Population and Public Health, University of British Columbia, Vancouver, Canada, 5 Department of International Health, Bloomberg School of Public Health, Johns Hopkins University, Baltimore, MD, United States of America

‡ JKM and JMN are co-first authors to this work.
* jill.murphy@ubc.ca

**Data Availability Statement:** All relevant data are contained with the paper and in the Web Annex to

## Abstract

### Introduction

This study explores the perspectives of workers and managers on workplace programs and interventions that seek to promote mental wellbeing, and prevent and treat mental health conditions The results contributed supporting evidence for the development of the WHO's first global guidelines for mental health and work, which provide evidence-based recommendations to support the implementation of workplace mental health programs and supports, to improve their acceptability, appropriateness, and uptake.

### Methods

An international online survey was used to examine the values and preferences among workers and managers related to workplace mental health prevention, protection, promotion, and support programs and services. The survey was made available in English, French, and Spanish and recruitment consisted of convenience sampling. Descriptive statistics were used to analyse the survey data. Rapid thematic qualitative analysis was used to analyse the results of open-ended questions.

### Results

N = 451 responses representing all WHO regions were included in the analysis. These results provide a unique international perspective on programs and supports for mental health at work, from the standpoint of workers and managers. Results suggest that workers value interventions developed in consultation with workers (including indicated, selective and universal interventions), increased training and capacity building among managers, and targeted interventions to address the pervasive impact of stigma on perceptions about mental health at work and help-seeking.

the WHO Guidelines on Mental Health and Work (https://apps.who.int/iris/bitstream/handle/10665/363102/9789240053076-eng.pdf).

**Funding:** This survey was commissioned by the World Health Organization to provide supporting evidence for the development of the WHO Guidelines on Mental Health at Work. The funding was held by the University of Alberta. JN and PAC were paid from these funds to support this research. WHO representatives participated in the design of the survey and supported its dissemination. Analysis was conducted independently by the study team and presented to WHO. A WHO representative reviewed this manuscript for consistency with WHO terminology but did not influence the findings or conclusions.

**Competing interests:** JKM, JN, PAC, GM and AG declared that no competing interests exist. EEM has received funding from Otsuka-Lundbeck for patient educational activities. RWL has received honoraria for ad hoc speaking or advising/consulting, or received research funds, from: Abbvie, Asia-Pacific Economic Cooperation, Bausch, BC Leading Edge Foundation, Brain Canada, Canadian Institutes of Health Research, Canadian Network for Mood and Anxiety Treatments, CAN-BIND Solutions, Carnot, Grand Challenges Canada, Healthy Minds Canada, Janssen, Lundbeck, Medscape, Michael Smith Foundation for Health Research, MITACS, Neurotorium, Ontario Brain Institute, Otsuka, Pfizer/Viatris, Shanghai Mental Health Center, Sunnybrook Health Sciences Centre, Unity Health, Vancouver Coastal Health Research Institute, and VGH-UBCH Foundation. This does not alter our adherence to PLOS ONE policies on sharing data and materials

## Conclusion

The findings of this study seek to reflect the perspectives of workers and their managers, and therein to promote improved access, availability and uptake of mental health programs and supports at work and–ultimately- to support the potential of workplaces as environments that promote and support mental health.

## Introduction and background

Mental health conditions, including substance-use disorders, are prevalent worldwide and lead to a substantial burden on individuals and high socioeconomic cost. Globally, mental health conditions are a leading cause of illness and disability [1, 2]. Left untreated, they are associated with poor physical health [3], reduced quality of life [4], lower functional capacity [5] and lost productivity [6]. People living with mental health conditions may also experience stigma, discrimination and human rights violations [7]. Despite the risks associated with poor mental health, approximately 50% of people worldwide do not have access to evidence-based mental health support, and in the lowest resourced countries this gap may be up to 90% [8].

Work and mental health are inextricably linked. Work may have a positive or negative impact on mental health, while mental ill health may affect work performance, productivity, and employment status [9]. Mental health conditions affecting workers include clinically diagnosed conditions like depression, anxiety or alcohol use disorders and sub-clinical issues including workplace stress and burnout [10]. Experiences of mental ill health may lead to negative impacts for workers including workplace discrimination [10]. The impact of untreated mental health conditions for workplaces include the costs of absenteeism (time off from work due to illness), presenteeism (reduced work capacity due to working while ill), lost productivity, increased risk of workplace injury or safety incidents, disability claims, and challenges with retention [11, 12]. It is estimated that in the world's 36 largest countries, the loss of workplace productivity due to untreated depression and anxiety is equivalent to 12 billion work days each year, costing up to US $925 billion [13].

People over the age of 18 years spend more than 60% of waking hours at work [14], suggesting that the workplace presents a considerable opportunity for the provision of mental health prevention, protection, promotion and support interventions. Considerable savings and financial benefit may result from investing in workplace mental health promotion and prevention [15]. Despite this, access to occupational health services remains low worldwide, with only 20–50% of workers in high income settings and 5–10% in low and middle-income countries (LMICs) able to access to such supports [16, 17].

Understanding the perspectives and experiences of workers is essential to informing how work and workplaces can best provide appropriate, targeted support to promote workforce mental health and wellbeing. The purpose of this global survey was to capture the perspectives of workers, including managers, in order to understand their values and preferences related to mental health at work. Specifically, we sought to 1) understand workers' perspectives and preferences related to different types of workplace programs and interventions to promote mental wellbeing and prevent and treat mental health conditions, including their help-seeking preferences, and 2) to assess workers' awareness of and access to specific types of mental health programs and supports at work. This survey was commissioned by the World Health Organization (WHO) to provide supporting evidence for the development of the WHO's Guidelines on mental health and work [18].

## Methods

### Survey scope and development

This survey examined values and preferences among workers and managers related to workplace mental health promotion, prevention and support programs and services. The survey development was led by co-first authors JN and JKM, in collaboration with co-author GM and the WHO technical officer responsible for the guideline development process. The survey scope and questions, including the intervention types that were included, were informed by the key questions identified for the Guideline Development Group and an Evidence to Decision Making Framework provided by the WHO partners. Survey questions were developed specifically for this survey based on a review of the relevant literature, and elements of the survey structure was informed by similar surveys including the WHO Consolidated Guideline on Self-care interventions for Health: Sexual and Reproductive Health and Rights Global Values and Preferences Survey [19]. In the Results section, we specify where survey items were informed by existing frameworks or classification systems.

The survey (see S1 File) explored the following themes: (1) Values and preferences related to mental health and work generally, including help-seeking preferences for programs and services, and; (2) awareness, access, and values and preferences related to specific intervention types. Specific intervention types are described in Fig 1. The survey consisted primarily of quantitative questions and we included short-answer, open-ended questions as part of the 'general values and preferences' section as described below.

The survey contained specific questions for managers, people who provide mental health support at work, and 'high-risk' workers including first responders, humanitarian and healthcare workers. When participants indicated that they were managers (indicating that they supervise employees), 'high-risk' workers (indicating that they work in a health, emergency or humanitarian services), and/or provide mental health support at work, the survey logic directed them to a specific set of questions designed for these participant subgroups, which

1. **Psychosocial/emotional** (e.g., talking to a professional, peer support worker or other care provider about feelings of stress, how to manage problems or symptoms of mental illness, or participating in self-help and coaching programs, etc.)
2. **Leisure-based physical activity** (e.g., an exercise routine that has been given to you by a doctor or another health professional)
3. **Health promotion** (e.g., combined programmes addressing a worker's lifestyle such as nutrition, physical activity, health education, stress management)
4. **Training/Education** to learn more about mental health, lowering stigma, or building skills to support the mental health and well-being of manager workers (incl. skills development and leadership coaching)
5. **Screening** to support mental health (identifying signs that someone is facing problems so that they can be supported early)
6. **Return to work** (after sickness absence related to mental health)
7. **Vocational support** to help people experiencing mental health problems to find and keep work
8. **Organizational changes** (e.g., organization policies or programs that lead to changes in the working environment, working conditions, work tasks for the purpose of supporting workers' mental health including workload and pace, work schedule, control over work, work environment and safety, organizational culture, relationships with managers and co-workers, career development, work-life balance).

**Fig 1. Mental health programs and supports.**

they responded to in addition to the questions that were available for all survey participants. Open-ended questions were included to provide a more comprehensive, qualitative understanding of attitudes related to mental health program and service provision at work in general, regarding the changing nature of work and mental health, the impact of the COVID-19 pandemic on work and mental health and attitudes related to mental health screening at work. For brevity, this paper focuses on the two themes identified above, with additional data available in the *WHO Web Annex*: *Evidence profiles and supporting evidence* (Available at: https://apps.who.int/iris/bitstream/handle/10665/363102/9789240053076-eng.pdf).

The survey focused on interventions delivered at or in relation to work that are provided at an individual, manager/ team, and organizational level. Interventions of interest include universal promotion and prevention programs (for the general worker population), selected promotion and prevention programs (for workers in high-risk occupations), indicated promotion and prevention programs (for workers showing signs of sub-threshold emotional distress), and programs to support treatment and recovery (for workers with symptoms of mental health or substance use conditions). The survey also examined perspectives on screening for mental health in the workplace. The full survey is available in the S1 File.

The survey was piloted by ten individuals representing a diversity of backgrounds and from several countries (Canada, the United States, Japan, Zambia, Peru, and the Netherlands), as well as several WHO-affiliated experts and expert groups. We asked pilot participants to provide feedback about their ability to understand the survey questions and terminology, the survey flow, appropriateness of the questions, and survey length. Pilot participant feedback was applied to the final survey, including removing repetitive questions, providing clear definitions of terminology, and rephrasing some questions for clarity.

Ethics approval for this study was obtained from the University of Alberta's Research Ethics Office (approval number: Pro00107550). Participants were required to provide written informed consent by selecting "I consent to participate in this survey" in order to access the survey. Participants were provided with the option to withdraw consent at any time while completing the survey by withdrawing from the survey before submission. All data from withdrawn surveys has been excluded from the final dataset.

### Survey recruitment, sampling and dissemination

This survey was disseminated online in English, French and Spanish using Qualtrics survey software [20]. Inclusion criteria for survey participation were employed civilian (non-military) adults over 18 years who are engaged in paid formal, non-standard work or informal work. The survey was open to people worldwide.

We used convenience sampling to generate survey participation. This consisted of broad dissemination via several channels, including through international global mental health networks, university networks, via mental health, workplace, and patient-serving organizations, and via social media. No incentives were provided to encourage survey participation. Data collection took place between April 9th and May 16th, 2021.

### Analysis

We used descriptive statistics to analyse the survey data. Responses to open-ended questions were analysed using rapid thematic qualitative analysis [21]. Lead author JKM conducted the thematic analysis, reviewing the responses for immersion in the data and developing a code book based on each question category, with new codes added as they were identified in the dataset. Data were coded using NVivo 12 [22] software, and reviewed to identify overarching themes related to each question category.

## Results

451 surveys were included in the final analysis after cleaning data to remove withdrawn surveys (indicating withdrawal of informed consent) and spam responses. Of these, 224 were complete and 227 were partially completed. We have included data from partially completed surveys except in cases where the survey respondents withdrew their consent. We have indicated in the results tables the total number (n) of responses for each question. In cases where participants could select more than one response ("choose all that apply"), we have indicated the total number of respondents and total number of response selections per item. 29 surveys were completed in Spanish, and 29 were completed in French, with the remaining 393 completed in English. Findings related to the general workforce and managers are reported here. Disaggregated findings related to workers in high-risk occupations will be reported elsewhere.

Quantitative and open-ended question results are described together where thematically aligned. For open-ended question responses, illustrative quotations are provided. Quotations are annotated with participant number based on the order in which they completed the survey, whether they are managers (i.e. supervise staff), and their gender.

### Participant demographics and types of work

451 individuals responded to the survey. Respondents were predominantly highly educated (64.2% with a graduate degree), identified as female (67.2%), averaged around 43 years old, and were living in large or medium sized cities (60.5%). 11.5% identified as sexual and gender minority groups, 18.6% identified as ethnic minority groups in their country of residence and 9.3% identified as Indigenous (Table 1).

Just over half (55.3%) of responses came from people residing in high-income countries, 21.4% from upper-middle income, 14.2% from lower-middle income and 8.9% from low-income countries. Survey responses came from all WHO regions, with a majority of responses from the European region (42.4%) and the region of the Americas (28.5%). Additional responses came from the Western Pacific (13.0%), African region (8.9%), the Eastern Mediterranean (4.8%), and South East Asia (2.4%).

Table 2 displays the employment characteristics of respondents, with the majority (58.3%) engaged in paid full-time formal employment.

Respondents selected the type of employment in which they are engaged by industry from a drop-down menu based on the United Nations' International Standard Industrial Classification of All Economic Activities (ISIC) [23]. They were able to further select more detail on their employment type via additional drop-down menus and by entering open field data. The majority of respondents are employed in Human Health and Social Work Activities (25.5%), Professional, Scientific and Technical Activities (21.5%) and Education (19.8%). More detail can be found in WHO Web Annex: Evidence profiles and supporting evidence, page. 307 (Available at: https://apps.who.int/iris/bitstream/handle/10665/363102/9789240053076-eng.pdf).

Over half of respondents indicated that they play a role in supporting mental health at work (53.3%) in roles including occupational health and safety, human resources and mental health workers. Just over half of respondents identified themselves as managers (51.9%), indicating that they play a supervisory role at work. A majority of respondents (68.1%) indicate that they currently or have previously experienced a mental health condition, and 28.1% have taken a formal leave due to mental health challenges.

**Theme 1: General values and preferences regarding mental health at work.**   This theme sought to explore respondent's general beliefs about mental health, and values and preferences regarding mental health programs and supports at work and help-seeking in general.

**Table 1. Participant characteristics.**

| Demographic Variable | Percent |
| --- | --- |
| Gender (n = 357) | Percent |
| Woman | 67.2 |
| Man | 31.4 |
| Other | 1.1 |
| Prefer not to say | 0.3 |
| Total | 100 |
| Median Age | 42.8 years |
| Community size (n = 348) | Percent |
| Large city (>1 million inhabitants) | 37.9 |
| Medium city (300,000–1 million inhabitants) | 25.6 |
| Small city (100,000–300,000 inhabitants) | 13.5 |
| Large town (20,000–100,000 inhabitants) | 10.3 |
| Medium town (1,000–20,000 inhabitants) | 8.3 |
| Small town, village or hamlet (<1000 inhabitants) | 4.3 |
| Total | 100 |
| Highest Level of Formal Education Completed (n = 321) | Percent |
| Graduate Degree (e.g., Master or Doctor Degree) | 64.2 |
| University Undergraduate Degree (e.g., Bachelor Degree) | 28.7 |
| Secondary/high school (including vocational high school) | 4 |
| Other | 2.8 |
| None | 0.3 |
| Primary school | 0 |
| Total | 100 |
| Identify as Member of Sexual or Gender Minority Group (n = 338) | Percent |
| No | 85.5 |
| Yes | 11.5 |
| Prefer not to say | 2.4 |
| Other | 0.6 |
| Total | 100% |
| Identify as Member of Racial or Ethnic Minority (n = 323) | Percent |
| No | 78.6 |
| Yes | 18.6 |
| Prefer not to say | 1.5 |
| Other | 1.2 |
| Total | 100 |
| Identify as Member of an Indigenous Group (n = 54) | Percent |
| No | 87.0 |
| Yes | 9.3 |
| Prefer not to say | 1.9 |
| Unsure | 1.9 |
| Total | 100 |
| Caregiving Responsibility at Home (n = 338) | Percent |
| No | 58.3 |
| Yes | 41.7 |
| Total | 100 |
| Identify as Living with a Disability (n = 314) | Percent |
| No | 87.3 |

(*Continued*)

**Table 1.** (Continued)

| Demographic Variable | Percent |
|---|---|
| Yes | 12.7 |
| Total | 100 |

## Current impact of work on mental health

To the question, "What type of impact does your work have generally on your mental health?" a majority (40.2%) of respondents reported that work had a positive impact on their mental health, with 24.5% reporting a somewhat positive impact and 15.7% reporting a very positive impact. 19.0% reported that work has a somewhat negative impact on their mental health, while 6.2% reported a very negative impact. 9.8% responded neutrally, and 0.7% were unsure.

We also asked respondents to rate on a six point Likert scale the extent to which several factors, based on established categories of psychosocial risk [24], negatively impact their mental health. As described in Table 3, issues related to "workload and work pace," including high workload, challenging deadlines and understaffed work environments, had the most negative impact on mental health.

We provided an open-response option, inviting participants to provide more context related to the impact of work on their mental health. Respondents provided a spectrum of responses, with some describing the positive impact of work on mental health, as described in the following quotation:

> *"My job contributes to positive well-being by providing pay, contact with other people, achievement and sense of importance/contribution to society. (P81, Worker, Man)"*

Others report a mixed impact of work on their mental health, with work contributing several positive factors but also contributing to stress:

> *"My work provides purpose and structure and social relationships which is important for my mental health. It also provides stress and uncertainty at times. There are positive aspects and negative aspects: it carries stress and burnout but also a sense of fulfillment. (P41, Worker, Woman)"*

Finally, other respondents describe a predominantly negative impact of work on mental health, often related to challenging relationships with management, overly demanding workloads and lack of work-life balance, and insufficient compensation:

**Table 2. Current employment characteristics.**

| Current Employment Status (n = 337) | *Percent* |
|---|---|
| Full-time employment | 58.3 |
| Self-employed/ small business owner | 8.4 |
| Part-time employment | 8.0 |
| Informal* | 3.1 |
| Other | 2.4 |
| Sickness leave/ disability leave | 1.3 |
| Casual, seasonal, or temporary | 1.1 |
| Family business | 0.4 |

*includes: Gig/ platform/ freelance/ contract work / zero hours contracts/ Crowdsource/ Casual, seasonal, or temporary/ informal

**Table 3. Psychosocial risk factors perceived negative impact on mental health.**

| (N = 291) | A great deal, % | A lot, % | A moderate amount, % | A little, % | Not at all, % |
|---|---|---|---|---|---|
| Workload and work pace (n = 264) | 20.8 | 19.3 | 27.7 | 17.8 | 13.3 |
| Organizational culture and function (n = 256) | 20.3 | 10.6 | 18.0 | 25.0 | 24.2 |
| Career development (n = 253) | 15.8 | 19.0 | 13.4 | 19.0 | 29.3 |
| Role in organization (n = 256) | 14.8 | 9.8 | 14.1 | 31.6 | 27.0 |
| Interpersonal relationships at work (n = 256) | 14.1 | 9.0 | 15.2 | 27.3 | 32.4 |
| Job control (n = 256) | 12.9 | 14.8 | 18.0 | 26.6 | 24.6 |
| Work schedule (n = 259) | 12.0 | 16.2 | 14.3 | 22.4 | 31.7 |
| Work content/ task-design (n = 286) | 11.5 | 11.2 | 18.2 | 19.6 | 36.7 |
| Work life balance (n = 253) | 11.1 | 16.2 | 18.2 | 24.5 | 28.1 |
| Environment and equipment (n = 255) | 8.6 | 9.4 | 10.6 | 16.9 | 46.7 |
| Job stability (n = 254) | 8.3 | 11.8 | 13.4 | 22.1 | 41.3 |

"*We are overworked with a lot of projects that we are responsible for, too many meetings, few opportunities for growth, feedback not is not acted upon by leadership/management, there is a lack of transparency about decisions and information, underpaid for value added (P 125, Worker, Woman)*"

## Help-seeking preferences

We assessed factors related to help-seeking preferences and accessibility of programs and supports among workers. Table 4 describes the comfort level of respondents when speaking about their mental health concerns at work with various individuals, workplace representatives and services. A majority report being most comfortable with speaking to health services outside of work, with 44.8% indicating they are very comfortable and 41.5% indicating they are comfortable. Participants reported being least comfortable speaking with human resources (23.1% uncomfortable and 17.9% very uncomfortable) and with managers (25.8% uncomfortable and 14.6% very uncomfortable).

We sought responses regarding whether respondents had considered participating in mental health programs and supports at work but did not do so and their reasons for not participating so as to elicit potential barriers to participation. 62.6% of respondents had considered participating in an intervention but chose not to. The majority (45.6%) report that they chose not to participate in psychosocial and emotional programs and supports. The reasons for not participating are reported in Table 5, with the most frequent reported reason being that they did not have time (17.5%), followed by a preference to manage mental health concerns themselves (13.5%) and concerns about privacy (13.5%).

**Table 4. Preferences for speaking about mental health concerns at work.**

| | Very comfortable % | Comfortable % | Neither comfortable or uncomfortable % | Uncomfortable % | Very uncomfortable % | N/A % |
|---|---|---|---|---|---|---|
| Health services outside of work (n = 212) | 44.8 | 41.5 | 9.0 | 3.8 | 0.5 | 0.5 |
| Colleagues (n = 231) | 19.9 | 37.6 | 16.7 | 19.0 | 5.9 | 0.9 |
| Health services at work (n = 211) | 19.0 | 30.3 | 16.1 | 10.0 | 5.2 | 19.4 |
| Managers (n = 213) | 14.1 | 28.2 | 16.4 | 25.8 | 14.6 | 0.9 |
| Union (n = 211) | 12.8 | 17.1 | 18.0 | 15.2 | 9.5 | 27.5 |
| Human resources (n = 212) | 12.7 | 24.1 | 19.3 | 23.1 | 17.9 | 2.8 |

**Table 5. Reasons for not participating in mental health programs or supports at work.**

| Reasons for Not Participating (n = 79; Total responses n = 222) | Percent |
|---|---|
| Did not have time (n = 39) | 17.5 |
| Preferred to manage by myself(n = 30) | 13.5 |
| Was worried about privacy (n = 25) | 11.3 |
| Was not able to access the services or program (n = 21) | 9.5 |
| I was worried that asking for help would lead to negative consequences (n = 17) | 7.7 |
| Did not think it would help/be effective (n = 17) | 7.7 |
| Afraid to ask for help from my manager or employer (n = 17) | 7.7 |
| Lack of choice in service types (n = 13) | 5.9 |
| Other, please specify* | 4.9 |
| Afraid my co-workers would find out (n = 10) | 4.5 |
| Did not feel I needed to participate (n = 9) | 4.0 |
| I am too focused on meeting my basic needs (food, clothing, shelter) (n = 6) | 2.7 |
| Could not afford it (n = 5) | 2.2 |
| Work is not responsible for supporting or promoting the mental health and wellbeing or workers (n = 2) | 0.9 |

*Other responses: Fear that management would respond negatively to need to take necessary time; feel that most representatives don't have necessary experience and belief that systems is not responsive to needs; Lack of trust in organizers of programs; Work would not provide referral to occupational health; Concerned that the service was not open for specific category of employment

As shown in Table 6, respondents also selected who they would prefer to receive support from across four scenarios related to experiencing mental health challenges at work. Respondents were able to select all that apply from a list of possible resources. As shown in Table 6, across all scenarios (first experiencing a mental health concern, seeking ongoing help for mental health concern, deciding whether to take a leave of absence or to return to work following a leave related to a mental health concern) participants rated managers or supervisors most highly, with responses ranging from 30.1% for returning to work after a leave to 25% for seeking ongoing support for their mental health at work. Union representatives were rated the lowest across all categories with responses ranging from 4% for first experiencing a mental health concern at work, to 7.1% for returning to work after a leave of absence.

When asked about the importance of accessing mental health supports on their own, without the assistance of involvement of a manager, a large majority (63.5%) of respondents indicated that doing so is very important, not wanting their employer to know they are accessing services or supports for mental health. Responses are displayed in Table 7.

**Table 6. Preferences for seeking support for mental health concerns at work.**

Who would you want support from when...

| | Colleagues | Managers/ supervisors | Human resources | Union reps | Health services at work | Health services outside of work |
|---|---|---|---|---|---|---|
| **First experiencing mental health concerns at work? (n = 218/ choice count n = 451)** | 22.4 | 26.4 | 8.0 | 4.0 | 15.7 | 23.5 |
| **Seeking ongoing support for mental health at work? (n = 49/ Choice count n = 92)** | 14.1 | 25.0 | 10.9 | 6.5 | 20.7 | 22.8 |
| **Returning to work after a leave for mental health reasons? (n = 44/ Choice count n = 113)** | 10.6 | 30.1 | 21.2 | 7.1 | 14.2 | 16.8 |
| **Deciding whether or not to take a leave of absence due to mental health? (n = 46/ Choice count n = 101)** | 5.9 | 28.7 | 18.8 | 5.0 | 16.8 | 24.8 |

**Table 7. Importance of accessing mental health supports without assistance of manager (n = 222).**

| | % |
|---|---|
| **Very important (I do not want my work to know I am accessing services)** | 63.5 |
| **Somewhat important** | 23.4 |
| **Neutral** | 4.5 |
| **Very unimportant (I do not mind if my employer knows that I am accessing services)** | 4.5 |
| **Unsure** | 1.8 |
| **Depends on the intervention** | 1.8 |
| **Somewhat unimportant** | 0.5 |

Respondents rated their preferences for how they prefer to access mental health programs if needed (including individual, group, in-person and digital options), where they prefer to access programs or services via the workplace and their preferred method of delivery or provider for mental health support, as shown in Table 8. Respondents showed a preference for accessing mental health programs outside of work across all categories, with 50.3% indicating they prefer to access programs on their own in person and 32.6% preferring to access them on their own online. Regarding location of programs, 42.1% prefer to access them outside of work but not at their home while 27.7% prefer to access them online. Finally, 53.0% prefer to seek support from a health worker.

**Table 8. Preferences for mode, location and provider of intervention delivery.**

| Mode (n = 210) | % |
|---|---|
| **On my own, in person** | 50.3 |
| **On my own, digitally (online/ self-directed)** | 32.6 |
| **In a group, in person** | 9.8 |
| **In a group, digitally (online)** | 5.8 |
| **Other[a]** | 1.5 |
| **Location (n = 219)** | **%** |
| **Outside of work (not home)** | 42.1 |
| **Online** | 27.7 |
| **At home** | 16.9 |
| **At work** | 12.3 |
| **I would not be interested in support** | 0.5 |
| **Other[b]** | 0.5 |
| **Provider (n = 218)** | **%** |
| **Health worker** | 53.0 |
| **Human resources/ occupational health manager** | 14.8 |
| **Online app or web based program** | 12.9 |
| **Manager** | 12.6 |
| **No one/ on your own** | 4.0 |
| **Other[c]** | 2.7 |
| **I would not be interested in support** | 0.0 |

[a] Not comfortable accessing mental health support at work; on my own over the phone

[b] Home is usually a safe space but during the COVID-19 pandemic this was not the case; the level of psychological safety of location is a priority; a neutral venue (not at work, the home, or an office due to concerns of abuse)

[c] Community-based support; telephone-based; vocational rehab consultant; family, friends, other peer counselors who don't work for my agency; counsellor

**Table 9. What is most important for employers to consider in offering a mental health service or program that is available to workers outside of work? (n = 206).**

| | % |
|---|---|
| Ensure anonymity in accessing services | 21.4 |
| Ensure it is evidence based and effective | 21.4 |
| Ensure no resulting negative consequences | 19.9 |
| Cost to worker | 8.7 |
| Provide time to ensure employees can access care | 7.3 |
| Easy to access and use (e.g. convenient, flexible times) | 6.3 |
| Accessible in multiple ways (e.g. online and in person) | 4.4 |
| Cost to the employer | 3.9 |
| Provide face to face intervention options | 3.4 |
| Provide engaging and interesting content | 2.4 |
| Ensure physical space between work and the service | 0.5 |
| Other | 0.5 |

We asked respondents to rank from most to least important considerations for employers when offering mental health supports or programs outside of the workplace. Table 9 displays the frequency with which each item was ranked as a top priority. Ensuring anonymity in accessing services and ensuring that services are evidence based were ranked as most important, with each receiving 21.4% of responses. The least important consideration was ensuring physical space exists between work and the service with 0.5% of responses, followed by providing engaging and interesting content (2.4%).

**Theme 2: Awareness of, access to, and values and preferences for specific intervention types.** In this section, the survey sought to explore respondents' awareness of and access to specific types of mental health programs and supports at work, as well as their values and preferences for those interventions for which they have familiarity. Respondents were asked to provide their perspective on the intervention types described in Fig 1.

## Awareness of services and program types

We asked respondents to select, from a list of intervention types, all program and service types that they had previously heard of. As shown in Table 10, awareness of service and program types ranges from approximately one half to approximately one third of respondents.

Table 11 describes whether respondents have access to each intervention type at work, whether they have previously used each intervention type, and, if not, whether they would be willing to.

**Table 10. Respondent awareness of intervention options for mental health at work.**

| Interventions (N = 248) | |
|---|---|
| Organizational interventions (N = 251) | 60.6 |
| Psychosocial interventions | 51.0 |
| Health promotion | 46.3 |
| Leisure-based physical activity | 45.0 |
| Training | 44.1 |
| Return to work | 38.8 |
| Screening programmes | 32.6 |
| Vocational support | 29.3 |

**Table 11. Access to, previous use and willingness to use programs or support at or through work.**

| Access to intervention: | | Access to: | Previously used: | Would you use it: |
|---|---|---|---|---|
| **Psychosocial interventions** | | **% (n = 227)** | **% (n = 219)** | **%(n = 120)** |
| | **Yes** | 71.8 | 41.5 | 69.2 |
| | **No** | 19.8 | 55.7 | 18.3 |
| | **Unsure** | 8.4 | 2.7 | 12.5 |
| **Leisure-based physical activity** | | **% (n = 190)** | **% (n = 182)** | **%(n = 99)** |
| | **Yes** | 42.1 | 40.1 | 67.7 |
| | **No** | 51.1 | 54.9 | 13.1 |
| | **Unsure** | 6.1 | 4.9 | 19.2 |
| **Health promotion** | | **% (n = 192)** | **% (n = 181)** | **%(n = 104)** |
| | **Yes** | 56.3 | 39.2 | 59.6 |
| | **No** | 32.3 | 47.5 | 21.2 |
| | **Unsure** | 11.5 | 3.3 | 19.2 |
| **Training** | | **% (n = 178)** | **% (n = 178)** | **%(n = 74)** |
| | **Yes** | 64.6 | 52.4 | 67.6 |
| | **No** | 28.7 | 45.8 | 21.6 |
| | **Unsure** | 6.7 | 1.8 | 10.8 |
| **Screening programmes** | | **% (n = 129)** | **% (n = 120)** | **%(n = 83)** |
| | **Yes** | 32.6 | 21.7 | 60.2 |
| | **No** | 47.3 | 71.7 | 20.5 |
| | **Unsure** | 20.2 | 6.7 | 19.3 |
| **Return to work** | | **% (n = 312)** | **% (n = 140)** | **%(n = 100)** |
| | **Yes** | 63.5 | 22.9 | 69.0 |
| | **No** | 20.9 | 72.1 | 10.0 |
| | **Unsure** | 15.5 | 5.0 | 21.0 |
| **Vocational support** | | **% (n = 129)** | **% (n = 106)** | **%(n = 87)** |
| | **Yes** | 30.3 | 13.2 | 56.3 |
| | **No** | 48.6 | 82.1 | 16.1 |
| | **Unsure** | 20.1 | 4.7 | 27.6 |
| **Organizational interventions** | | **% (n = 133)** | **% (n = 129)** | **%(n = 60)** |
| | **Yes** | 49.6 | 45.7 | 80.0 |
| | **No** | 33.1 | 46.5 | 13.3 |
| | **Unsure** | 17.3 | 7.8 | 6.7 |

We asked respondents who have previously used programs and supports at work to rate the ease of accessing each intervention type. For respondents who had not previously accessed these interventions, we asked them to rate how easy they perceive access to be. As shown in Table 12, respondents who had previously accessed programs and supports at work reported easier access compared with perceptions about ease of access by those who had not accessed interventions:

When asked to rate the level of importance of specific intervention types in supporting mental health at work, the majority of respondents selected extremely or very important across all categories, as described in Table 13.

## Benefits and concerns

*Benefits by intervention type.* We assessed perceived benefits and concerns by intervention type as shown in Tables 14 and 15. We asked respondents to rate perceived benefits by intervention

**Table 12. Actual and perceived ease of access.**

| Easy to access intervention? | | Ease of access (if previously used): | Perceived ease of access (if previously not used): |
|---|---|---|---|
| **Psychosocial interventions** | | **% (n = 91)** | **% (n = 122)** |
| | **Very easy** | 28.5 | 19.7 |
| | **Easy** | 47.6 | 36.9 |
| | **Neutral** | 13.2 | 20.5 |
| | **Difficult** | 8.8 | 8.2 |
| | **Very difficult** | 1.1 | 4.9 |
| | **Unsure** | 1.1 | 9.8 |
| **Leisure-based physical activity** | | **% (n = 72)** | **% (n = 100)** |
| | **Very easy** | 40.2 | 20.0 |
| | **Easy** | 36.1 | 23.0 |
| | **Neutral** | 13.9 | 27.0 |
| | **Difficult** | 8.3 | 11.0 |
| | **Very difficult** | 0.0 | 4.0 |
| | **Unsure** | 1.4 | 15.0 |
| **Health promotion** | | **% (n = 71)** | **% (n = 104)** |
| | **Very easy** | 32.4 | 8.6 |
| | **Easy** | 53.5 | 22.1 |
| | **Neutral** | 7.0 | 30.8 |
| | **Difficult** | 4.2 | 10.6 |
| | **Very difficult** | 2.8 | 8.6 |
| | **Unsure** | 0.0 | 19.2 |
| **Training** | | **% (n = 87)** | **% (n = 73)** |
| | **Very easy** | 35.6 | 9.6 |
| | **Easy** | 41.4 | 24.7 |
| | **Neutral** | 20.7 | 27.4 |
| | **Difficult** | 1.1 | 13.7 |
| | **Very difficult** | 1.1 | 4.1 |
| | **Unsure** | 0.0 | 20.5 |
| **Screening programmes** | | **% (n = 26)** | **% (n = 86)** |
| | **Very easy** | 38.5 | 2.3 |
| | **Easy** | 34.6 | 19.8 |
| | **Neutral** | 7.7 | 26.7 |
| | **Difficult** | 11.5 | 19.9 |
| | **Very difficult** | 3.8 | 17.4 |
| | **Unsure** | 3.8 | 19.8 |
| **Return to work** | | **% (n = 32)** | **% (n = 100)** |
| | **Very easy** | 15.0 | 16.0 |
| | **Easy** | 21.9 | 23.0 |
| | **Neutral** | 9.4 | 21.0 |
| | **Difficult** | 12.5 | 14.0 |
| | **Very difficult** | 6.2 | 7.0 |
| | **Unsure** | 0.0 | 19.0 |

*(Continued)*

**Table 12.** (Continued)

| Easy to access intervention? | | Ease of access (if previously used): | Perceived ease of access (if previously not used): |
|---|---|---|---|
| **Vocational support** | | **% (n = 14)** | **% (n = 86)** |
| | **Very easy** | 21.4 | 2.3 |
| | **Easy** | 28.6 | 20.9 |
| | **Neutral** | 21.4 | 18.6 |
| | **Difficult** | 14.3 | 15.1 |
| | **Very difficult** | 7.1 | 10.5 |
| | **Unsure** | 7.1 | 32.6 |
| **Organizational interventions** | | **% (n = 59)** | **% (n = 70)** |
| | **Extremely easy** | *16.9* | 5.7 |
| | **Somewhat easy** | 35.6 | 20.0 |
| | **Neither easy nor difficult** | *16.9* | 18.6 |
| | **Somewhat difficult** | 25.4 | 21.4 |
| | **Extremely difficult** | 4.1 | 17.1 |
| | **Unsure** | NA | 17.1 |

type according to the following criteria: convenience, being helpful in promoting or supporting mental health, removing barriers like stigma, affordability for the worker, affordability for the company, and benefiting the morale of the organization. As described in Table 14, an intervention being helpful for the worker's mental health was the most highly rated benefit across all intervention types, followed by the perceived convenience of an intervention, affordability for the worker, and being beneficial to the whole organization. For the training and education intervention, the potential to reduce stigma was also rated highly by 19.7% of respondents.

*Benefits of supporting mental health at work*. Responses to open-ended questions describing the benefits of promoting and supporting mental health at work, perceived benefits for workers included improved overall quality of life, well-being, and job satisfaction. It was noted that workers may be empowered and "flourish" through the establishment of a healthy and supportive work environment, as described in the following quotation:

> "*Ppl [sic] spend the bulk of their waking hours in work, so I believe that organizations have a duty to go beyond simply keeping places safe and doing no harm, but that by addressing primary concerns at the organizational level employees can thrive. That should be the goal!*"
> (P267, Worker, Woman)

**Table 13. To what extent do respondents value the potential interventions for mental health and work?**

| Interventions (N = 275) | Extremely important % | Very important % | Moderately important % | Slightly or not at all important % |
|---|---|---|---|---|
| **Manager training for mental health** | 62.4 | 29.7 | 4.5 | 3.4 |
| **Organizational interventions** | 61.4 | 30.7 | 4.9 | 3.0 |
| **Return to work** | 60.1 | 33.1 | 3.8 | 3.0 |
| **Support for workers with mental health problems (e.g. work accommodations)** | 58.5 | 35.1 | 3.0 | 3.4 |
| **Access to mental health promotion and prevention (e.g. individual psychosocial interventions, leisure-based physical activity)** | 52.0 | 36.3 | 8.1 | 3.7 |
| **Screening programmes** | 42.2 | 35.4 | 15.6 | 6.8 |
| **Vocational support** | 40.8 | 34.4 | 18.7 | 6.1 |

**Table 14. Perceived benefits of interventions.**

| Benefits: | Psychosocial interventions (Total respondents n = 225; Total responses n = 714) % | Leisure-based physical activity % (Total respondents n = 188; Total responses n = 566) | Health promotion% (Total respondents n = 190; Total responses n = 562) | Training % (Total respondents n = 175; Total responses n = 588) | Screening programmes % (Total respondents n = 125; Total responses n = 309) | Return to work % (Total respondents n = 144; Total responses n = 407) | Vocational support % (Total respondents n = 107; Total responses n = 278) | Organizational interventions % (Total respondents n = 133; Total responses n = 468) |
|---|---|---|---|---|---|---|---|---|
| Benefits mental health | 25.4 | 28.3 | 26.2 | 23.1 | 28.2 | 29.0 | 25.9 | 23.9 |
| Affordable for worker | 19.8 | 19.6 | 18.0 | 14.3 | 15.5 | 15.5 | 15.8 | 14.2 |
| Convenient | 18.9 | 19.4 | 19.6 | 15.8 | 16.5 | 17.0 | 16.6 | 14.0 |
| Benefits morale | 15.3 | 16.3 | 16.2 | 19.9 | 14.2 | 15.0 | 14.0 | 22.4 |
| Removes barriers (i.e. stigma) | 13.3 | 5.7 | 9.6 | 19.7 | 10.0 | 11.6 | 13.3 | 14.6 |
| Affordable for employer | 5.6 | 7.2 | 6.9 | 5.6 | 6.5 | 7.1 | 5.0 | 11.8 |
| Other[a] | 1.3 | 1.8 | 0.9 | 0.5 | 1.3 | 1.5 | 0.7 | 1.3 |
| There are no benefits | 0.3 | 0.2 | 0.2 | 0.2 | 0.7 | 0.3 | 0.0 | 0.0 |
| Unsure | 0.3 | 1.6 | 2.5 | 0.9 | 7.1 | 3.2 | 8.6 | 1.6 |

[a]Other e.g.: prevents illness; convenient for employer; demonstrates workplace commitment to health; intervention options can be helpful but addressing risk factors for mental health is also important.

Supporting mental health programs and services at work was also noted as a contributing factor for improving mental health literacy, and, through the provision of information on pathways to care and support, may also improve timely access to care and promote help-seeking among workers. Stigma reduction is also described as a benefit of promoting mental health at work by creating workplace cultures that are open, accepting and are adequately equipped to provide appropriate supports.

*Concerns by intervention type.* We also asked respondents to rate their concerns by intervention type according to the following criteria: lack of privacy and confidentiality, fear of judgement or stigma, inconvenience, difficulty in accessing it, being expensive for workers, and being expensive for the company. Regarding key concerns (Table 15) related to each intervention type, a majority of respondents report no concerns related to physical exercise interventions (34.2%), training and education programs (30.4%) and health promotion programs (29.4%). Across all other intervention types, fear of judgement or stigma, and privacy and confidentiality are rated as the highest concerns.

*Concerns with supporting mental health at work.* Similar concerns were raised in the responses to open-ended questions, where respondents were asked to describe any worries they had with regard to promoting and supporting mental health at work, and stigma was raised as a key theme. More specifically, respondents worried that stigma related to mental health would prevent workers from disclosing their mental health challenges and might lead to discrimination or other negative impacts in the workplace.

Ongoing stigma, often associated with low mental health awareness, was also described as a concern. Respondents cautioned that this lack of understanding and awareness may lead to

**Table 15. Perceived concerns of the interventions.**

| Concerns: | Psychosocial interventions (Total respondents n = 222; Total responses n = 417) % | Leisure-based physical activity % (Total respondents n = 185; Total responses n = 240) | Health promotion % (Total respondents n = 189; Total responses n = 265) | Training % (Total respondents n = 172; Total responses n = 237) | Screening programmes % (Total respondents n = 124; Total responses n = 239) | Return to work % (Total respondents n = 143; Total responses n = 253) | Vocational support % (Total respondents n = 106; Total responses n = 185) | Organizational interventions % (Total respondents n = 130; Total responses n = 197) |
|---|---|---|---|---|---|---|---|---|
| Fear of judgment/ stigma | 31.7 | 10.0 | 11.3 | 13.5 | 34.7 | 31.6 | 27.0 | 18.3 |
| Lack of privacy/ confidentiality | 24.0 | 6.3 | 14.0 | 10.1 | 27.6 | 25.7 | 20.0 | 14.2 |
| No concerns | 9.4 | 34.2 | 29.4 | 30.4 | 7.5 | 11.5 | 10.3 | 15.7 |
| Difficult to access | 8.4 | 10.0 | 8.7 | 10.1 | 8.4 | 10.7 | 15.7 | 10.7 |
| Expensive for worker | 7.0 | 7.1 | 8.3 | 7.2 | 4.2 | 3.2 | 5.4 | 3.5 |
| Other[a] | 7.0 | 9.2 | 7.9 | 7.6 | 4.6 | 5.1 | 2.7 | 8.6 |
| Inconvenient | 6.7 | 11.3 | 7.6 | 9.7 | 5.9 | 5.5 | 4.9 | 8.1 |
| Expensive for employer | 5.3 | 2.9 | 4.9 | 6.8 | 3.8 | 2.8 | 3.8 | 15.7 |
| Unsure | 0.7 | 9.2 | 7.9 | 4.6 | 3.4 | 4.0 | 10.3 | 5.1 |

poor treatment of people who are open about their mental health challenges at work, as described below:

> "*Because of the stigma around mental health, the people who need help the most are probably the people who are not going to reach out for help. Colleagues and managers should know what to look out for and how to approach someone who they think may be suffering a mental health issue. Seeking help for, and offering assistance to someone who is struggling with a mental health problem, should become as common placed [sic] as seeking help and assisting someone with a physical health problem*" (P285, Worker, Woman).

Of additional note, concern related to stigma was highest for screening interventions, with 34.7% of respondents indicating concerns about judgement and stigma and 27.6% reporting concerns with privacy and confidentiality (Table 15).

The quality of mental health programs and supports at work is also a concern among respondents, as illustrated in the following quotation that describes reluctance among workers to seek out existing mental health supports:

> "*. . .a lot of employers like higher education institutions rely on commercial and short-term mental health services that use a 'cookie cutter' approach to CBT and resilience. Staff have noted that these services are not very beneficial in the long run and have learnt not to seek them out.*" (P392, Worker, Man)

Respondents also emphasized the need for programs and supports that are appropriate and responsive to the needs of workers, calling for collaborative approaches to develop policies and practices that support their mental wellbeing, including by engaging workers with lived experience of mental health conditions, to understand how workplaces can fully support a range of mental health needs.

"*I believe the best mental health initiatives are done in collaboration with workers and their representatives. Managerial approaches to promoting mental health sometimes don't work so well, but a collaborative approach is best to ensure organizational cultural change.*" (P276, Worker, Man)

*Benefits and concerns related to mental health screening at work.* Respondents were asked to reflect on the potential benefits and harms specifically around mental health screening at work via a short-answer question. In response, participants described several concerns, including privacy, confidentiality and the use of results or data by employers or other parties, including insurance companies. Stigma and discrimination as a result of mental health screening was also raised as a concern with some stating that workers would be reluctant to participate in screening for fear of stigma, ostracization or negative repercussions related to their role at work. Ethical considerations of screening for mental health at work were also raised when pathways to care or services may be limited or unavailable. Some participants questioned the legal implications around duty of care for an organization conducting mental health screening.

Many respondents questioned the appropriateness of the workplace for conducting mental health screening and the validity of an employer having this information about worker mental health, as described below:

"I don't believe that the workplace is the setting in which screening should take place. There is a lot of risk for stigma as a result and it is only in very specific circumstances where an employer would need to know the details of this. Workplaces can be a conduit of information, but I would be concerned about them getting 'too close' to the actual screening." (P267, Worker, Woman)

Similarly, concerns regarding punitive measures associated with mental health screening at work were also raised by many respondents, as shown in the quotation below:

"*There is evidence that those who share are penalised either by withdrawing promotion, actual demotion or bullied. The anonymity of reporting has to be fiercely guarded.*" (P150, Worker, Woman)

Related to these concerns is the suggestion that screening and subsequent interventions or supports should take a population-based approach, avoiding the potential to single out and potentially marginalize individual workers:

"Screening needs to be done with a view to supporting all workers, and working out who needs what specific additional support. It should not be done as a way of weeding people out, or carrying negative connotations. For example, it would be beneficial to offer mental health awareness and a voluntary stress management training programme. It would be potentially harmful to single out staff members and tell them to complete a stress management programme." (P285, Manager, Man)

Despite the concerns raised regarding screening for mental health at work, positive themes were identified as well. Possible perceived benefits include the potential for mental health screening at work to lead to prevention and early identification of mental health conditions, thus leading to earlier intervention or care access. Shifts towards a workplace culture that is more supportive and open to discussing issues of mental health and wellbeing was another

potential benefit. Proper training and additional supports in the workplace can lead to a workplace culture that actively promotes mental wellbeing were also identified, as described below:

> "*Managers should be trained on the sensitivity of mental health information. When an organisation gets it right, like some do, the psychological safety spills into other aspects of work and life of those affected and related.*" (P150, Worker, Woman)

## Organizational changes

Respondents indicated their awareness of, experiences with, and perceived benefits and concerns about policies or programs aiming to support worker mental health via organizational changes such as those that lead to changes in the working environment, working conditions or work tasks for the purposes of supporting workers' mental health. We highlight these responses separately as the question types differed slightly from those related to the seven interventions described above.

When asked about perceived benefits of policies and programs to support worker mental health via organizational changes (Table 14) the greatest perceived benefits include being helpful to workers' mental health (23.9%) and benefitting the whole organization (23.9%).

As extracted from responses to open-ended questions, perceived organizational benefits include improved work quality and productivity among workers, improved workplace culture, greater retention, and reduced absenteeism and presenteeism. Also noted was the anticipated high potential return on investment for businesses or organizations investing in mental health and wellbeing, as described below:

> "*A healthy (in a broad sense) workforce is generally a more productive workforce—money invested up-front on preventive strategies (education, stigma reduction, management training), early intervention (HR policies), health care and return to work programs reap the benefits later in numerous ways.*" (P275, Manager, Man)

Regarding concerns about introducing policies and programs to support worker mental health via organizational changes (Table 15), the greatest concern among respondents was fear of judgement or stigma (18.3%), followed by cost for the company (15.7%).

**Table 16. What would help managers to feel more confident to support workers in distress/experiencing mental health issues or challenges.**

| | | % |
|---|---|---|
| *Confidence to support a worker in distress /experiencing mental health problems (n = 106)* | Somewhat confident | 40.6 |
| | Very confident | 38.7 |
| | Neutral | 10.4 |
| | Somewhat unconfident | 7.6 |
| | Very unconfident | 2.8 |
| *What would help managers feel more confident to support their workers? (n = 121)* | (More) training | 26.5 |
| | (More) information on interventions | 26.5 |
| | Need organizational infrastructure | 23.1 |
| | (More) information on mental health | 16.5 |
| | I feel confident and informed | 5.0 |
| | Other | 2.5 |

Responses to open-ended questions show a clear desire among respondents for upstream interventions and organizational shifts to support mentally healthy workplaces and to address factors in the workplace that may contribute to poor mental wellbeing. Similarly, respondents raised concerns about organizations creating policies related to workplace mental health without investing in meaningful change to address mental health risk factors. One respondent stated: *"Policies aren't worth the paper they are printed on without good practice."* (P273, Manager, Woman). Another cautioned against over pathologizing the workforce by equating challenges with mental health conditions or poor mental wellbeing instead of with the work environment, thus shifting responsibility to the worker instead of the employer:

> "*I am concerned that current efforts to support workers mental health focus on building resilience and maximizing worker productivity, rather than putting in systems that ensure better work-life balance.*" (P392, Worker, Man)

## Perspectives of managers

The survey assessed managers' perspectives and experiences with providing mental health programs and supports at work. All respondents who indicated that they supervise staff are considered to be managers and were directed to a suite of questions to understand their experience with providing mental health support at work, including via information and referrals and Employee Assistance Programs (EAPs), their perceived self-efficacy regarding providing mental health support to workers, and capacity building needs related to providing mental health support at work.

We asked managers if they have previously provided a referral or information to workers for where or how to access mental health support. 67.3% indicate that they have previously provided a referral or information to workers for where or how to access mental health support. Reasons for not previously having done so include, "It is not related to my job" (46.9%), "It is not available at/through my work" (21.9%), "I do not know what is available at work" (6.3%), or "other" (25%). For 'other', reasons provided by managers include not being approached by a worker for support, never having perceived a need among workers for mental health support, and not feeling they know their employees well enough to approach the topic of mental health.

Manager respondents report a high level of confidence to support workers experiencing psychological distress or experiencing mental health conditions, with 38.7% indicating they are very confident and 40.6% stating they are somewhat confident. Despite high levels of confidence selected by managers, when asked to indicate which resources would help to increase their confidence, only 5.0% indicate that they feel sufficiently confident and well-informed (Respondents who indicated they felt 'very confident' in the previous question were not directed to this question.). Table 16 describes factors that would help to improve manager confidence to support mental health among workers.

We also asked managers to identify how easy it would be for them to access training to support knowledge, skills, attitudes, and behaviour to improve mental health of workers when needed. 60.0% indicate that they can access this training when needed, 30.5% have no access to training and 9.5% selected "other". 'Other' responses included some respondents who were unsure about access to training, some managers indicating they can access training but only sporadically, and some stating that training is not yet offered but will be in the future.

Finally, we asked managers to report whether they have experience with EAPs, including through training, developing or managing these programs. A large majority (71.7%) do not have any experience with EAPs, while 28.3% report previous experience.

### Intervention specific

Managers have most frequently provided a referral to or information about psychosocial and emotional programs and supports (67.8%) and training and education programs (61.4%). They are least likely to have provided a referral to or information about vocational programs, with 40.0% indicating they had not done so, 33.3% indicating that they had, and 13.0% indicating these programs are not available where they work. Screening was also rated low, with 39.2% indicating they had not, 33.8% indicated they had and 17.6% indicating screening programs are not available where they work. With the exception of screening programs (33.8%) and vocational support programs (26.8%), approximately half of managers indicate that they feel very confident and informed about each intervention type. Managers report the highest level of confidence about psychosocial and emotional supports and programs at 59.2%. Approximately one quarter of respondents indicate the need for more information about each type of intervention, although the need for more information and training is indicated for vocational support (more information: 39.3%; more training: 23.2%), return to work programs (more information: 30.4%; more training: 15.2%) and screening programs (more information: 32.5%; more training: 20.0%). We also captured manager experiences and perspective with policies and programs to support work mental health through organizational changes. 45.2% of managers have provided information to employees about accessing or participating in such an intervention, while 28.8% have not and 17.8% indicate that these policies and programs are unavailable in their workplace.

## Discussion

The results of this survey provide a comprehensive understanding of the values and preferences of workers and managers related to the provision of mental health programs and supports at work. Below we present key considerations in reflection of survey findings.

### Organizational factors

The survey results show that workers place a high level of importance on organizational factors that may either contribute to negative mental health at work or promote mental wellbeing and psychological safety in the workplace. High workload and challenging workflow and organizational culture were most highly rated as having a negative impact on mental health, with open-ended response results emphasising the extent of these concerns among survey respondents. This suggests a need for employers and management to improve overall support for workers including via upstream interventions that support factors like work-life balance, healthy interpersonal relationships at work, prevention of extreme overwork, and effective communication between management and workers. These findings are consistent with research that describes the importance of prevention via workplace policies as an essential component of supporting mental health at work [10]. Several studies have found that organizational interventions can be beneficial for worker mental health, including for reducing burnout [25–27], though more research is needed regarding the implementation of these intervention in diverse contexts [27].

### Training and capacity building

Responses by both general workers and managers indicated that more training and capacity building for managers is necessary for providing effective mental health support at work. Although the majority of managers indicate they have some level of confidence to support worker mental health, 95% identified capacity building such as enhanced training and

information about mental health issues and interventions as necessary to improve their ability and confidence to provide mental health support to workers. Almost one half of managers report that they had not provided information or referrals for mental health programs and supports because they don't perceive it to be part of their job. Others indicate that they have never done so because no worker has ever reached out to them for support. While workplace policies regarding assessing and monitoring psychological safety and mental health vary by region and country, these findings suggest there is a need for enhanced training and capacity building for managers, including by supporting them to take a proactive role in identifying and supporting employee mental health. Evidence supports the importance of management training to promote mental health at work [28]. For example, an Australian study found that implementing a mental training program for managers working with fire and rescue workers led to a significant reduction in sick leave by workers and a return on investment of the equivalent of almost £10.00 per pound spent on training [29].

## Stigma

The issue of stigma and related concerns, including experiences of discrimination or punitive measures related to discussing or disclosing mental health concerns, was pervasive throughout the survey results. When describing general values and preferences related to mental health at work, perspectives on specific intervention types, and help-seeking preferences, stigma is at the forefront of concern among workers. Stigma-related issues emerged in several ways. When asked to rate their concerns related to specific intervention types, respondents rated worry about experiencing judgement and stigma and about privacy and confidentiality as top concerns across four intervention types: psychosocial and emotional, screening, return to work, and vocational support programs. Notably, the three interventions for which respondents indicated they had no concerns are interventions that can be delivered at the population level and do not involve singling out specific individuals due to mental health-related concerns: leisure-based physical activity, health promotion, and training and education.

Concerns about stigma are also evident in responses related to help-seeking preferences. Responses regarding preference for sharing mental health concerns suggest a high degree of reticence to openly seek mental health support at work. Respondents prefer to access care from external health services and are most concerned with ensuring that they are able to access programs and supports anonymously and without experiencing negative repercussions at work. This is also reflected in preferences for how to access care- most prefer to access care alone and in-person, and just over one quarter prefer to access support online via digital options. Respondents also report their preference for talking about mental health concerns outside of work or with health services at work, and report high levels of discomfort with speaking with human resource representatives or management.

The substantial influence of stigma and related concerns are consistent with the findings of an expert survey conducted in Europe and Australia, which found that workers are frequently reluctant to speak openly about mental health [30]. Our findings show that concerns about experiencing discrimination and other negative repercussions related to being open about or seeking support for mental health at work suggests the potential value of targeted anti-stigma campaigns and interventions to promote a culture of openness and promote help-seeking at work. Evidence suggests that anti-stigma campaigns can improve comfort with disclosing mental health issues, including with employers [31] and can improve supportive behaviour among workers [32, 33]. Interventions based on social contact among people without and with lived experience of mental health condition are identified as highly effective in reducing mental health stigma [34]. Related to the need for organizational change and supports to promote

mental health and wellbeing, enforced policies that protect workers against the negative impacts of stigma related to mental health at work may also be warranted.

## Screening

Respondents show a high level of ambivalence about the use of screening for mental health at work. Quantitative results demonstrate low levels of knowledge, awareness and acceptance of screening programs at work compared with other intervention types. Survey respondents rated screening programs most highly compared with other intervention types for concerns about stigma and judgement, and screening programs are the least recommended by respondents for implementation at work. The responses to open-ended questions further demonstrate a high level of concern regarding the use of mental health screening at work. Respondents did, however, describe potential positive impacts of screening for mental health at work when done carefully and at a population level, including facilitating early detection and intervention. A US study testing a population-level, opt-in screening intervention with the option to access follow-up care found that the screening program facilitated care uptake among workers [35]. However, a 2023 systematic review and meta-analysis [36] found minimal evidence to support workplace mental health screenings and found that screening programs when implemented alone do not support improved mental health. When implemented on a voluntary basis at the population level with adequate privacy and confidentiality measures and available support for those who need it, screening programs may be beneficial in improving early detection and promoting pathways to care, but should not be relied on alone to improve mental health outcomes.

## Worker preferences

The survey results provide several key insights into respondents' preferences regarding specific intervention types. Primarily, in addition to screening programs, respondents report the lowest levels of knowledge, experience and access to return to work and vocational support programs. Manager responses also indicate a low level of experience with and capacity related to these types of interventions. As long-term sickness absence and unemployment are themselves possible consequences of untreated mental health conditions [37], these findings suggest that improved investment in and awareness about such programs and supports among workers and managers would be beneficial.

## Barriers to access

For those who have not previously accessed mental health programs or supports at work, responses related to perceived ease of access, for which a majority of respondents selected 'neutral' or 'unsure/don't know' responses across all intervention types, are low compared with responses from those who have actually accessed these supports. This indicates low levels of knowledge about the availability and accessibility of programs and supports. Previous evidence shows that uptake of workplace mental health programs is often low among workers [38]. A Canadian study that implemented mental health awareness training for managers found that improving manager awareness about mental health and mental health interventions increased manager support for workers experiencing mental health conditions and improved worker help-seeking and uptake of mental health supports [39].

Finally, when asked which factors had deterred them from accessing mental health programs and supports at work, respondents indicated that time was the biggest access barrier. Time constraints include workload demands making it difficult for workers to access support, and lack of flexibility in how programs are delivered (e.g. after work hours). Time has

previously been identified as a structural barrier to mental health service use by workers experiencing depression [40]. Allowing flexibility in work hours to enable workers to participate in mental health programs and supports may thus support increased access to programs and supports.

### Including people with lived experience of mental health conditions

The results of the open-ended survey questions include a strong indication of the need for participatory approaches to designing and implementing mental health supports at work, including by involving people with lived or living experience of mental health conditions in the development of organizational policies and programs. The involvement of service users in design and improvement of mental health services is increasingly recognized as essential in the broader mental health field [41]. For example, a qualitative study that elicited barriers and facilitators to use of a digital intervention to support workplace mental health [42] points to the importance of engaging with service users in intervention design and implementation planning. This is particularly important when working with multicultural or otherwise diverse teams or when adapting interventions from one setting to another.

### Limitations

The survey sample includes a majority of respondents that are female, reside in Europe and the Americas, are urban-dwelling, highly educated, employed full-time with large businesses or organizations, and have access to benefits (e.g. sick days, vacation time) through their place of work. Over half of respondents work in a role that involves supporting mental health at work, suggesting they are highly familiar with the survey subject matter. Using convenience sampling can bias responses towards people with a pre-existing interest in the topic and with sufficient time and literacy to complete an extensive survey. The survey was available in three languages (English, French and Spanish), which may have limited participation from respondents who do not speak these languages. We believe that this survey offers a comprehensive overview of values and preferences for mental health programs and supports at work among workers internationally, but further research to capture more diverse perspectives could further enrich the findings. Almost 70% of survey respondents have experienced a mental health condition, indicating that this survey has captured the perspectives of people with lived experience of mental health concerns in the workplace. It also reinforces the need for enhanced mental health support at work.

### Conclusions

The results of this study underscore the importance of understanding the perspectives of workers and managers to inform the development and implementation of mental health programs and supports that are acceptable and appropriate for workers themselves, helping to improve uptake and to inform strategies for implementation. The results indicate the need for both population-level and targeted supports that are developed collaboratively with workers, for increased training and capacity building among managers, and for targeted interventions to address the pervasive impact of stigma on perceptions about mental health at work and help-seeking. These results provide a unique and global perspective related to mental health at work from the perspective of workers, managers, and providers of mental health supports at work. The findings directly informed the development of the WHO Guidelines on Mental Health at Work [18] which were launched in the fall of 2022 and will help to advance workplace mental health support globally.

## Supporting information

**S1 File. Mental health at work survey.**
(PDF)

## Acknowledgments

The authors would like to sincerely thank everyone who participated in the survey for sharing their time and expertise.

## Author Contributions

**Conceptualization:** Jill K. Murphy, Jasmine M. Noble, Georgia Michlig.

**Data curation:** Jill K. Murphy, Jasmine M. Noble.

**Formal analysis:** Jill K. Murphy, Jasmine M. Noble, Promit Ananyo Chakraborty.

**Funding acquisition:** Jill K. Murphy, Erin E. Michalak, Andrew J. Greenshaw, Raymond W. Lam.

**Investigation:** Jill K. Murphy.

**Methodology:** Jill K. Murphy, Jasmine M. Noble.

**Project administration:** Jill K. Murphy, Jasmine M. Noble.

**Supervision:** Jill K. Murphy, Raymond W. Lam.

**Writing – original draft:** Jill K. Murphy, Jasmine M. Noble.

**Writing – review & editing:** Georgia Michlig, Erin E. Michalak, Andrew J. Greenshaw, Raymond W. Lam.

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
