## [Decision Letter · Decision Letter 0]

28 Jun 2023

PONE-D-23-05532Values and Preferences Related to Workplace Mental Health Programs and Interventions: An International SurveyPLOS ONE

Dear Dr. Murphy,

Thank you for submitting your manuscript to PLOS ONE. After careful consideration, we feel that it has merit but does not fully meet PLOS ONE’s publication criteria as it currently stands. Therefore, we invite you to submit a revised version of the manuscript that addresses the points raised during the review process.

 **Provide more clarity regarding the  research question and the study objectives in the introductory sections including the Abstract. ****Provide more information in the abstract regarding the survey, including sample size, etc.****Clarify whether this was quantitative survey with a few open-ended questions, rather than a mixed methods study using both quantitative and qualitative methods****Clarify whether study participants represent employees/ supervisors, rather than employees versus employers?****Clarify whether the study employed a validated questionnaire or whether the questionnaire was developed exclusively for this study? Was the study instrument validated, if so how? **** Consider submitting a copy of the study Questionnaire as a supplementary file if not already included in your supplementary materials?****Expand or extend the Results and Discussion sections to provide more details and clarity for readers as suggested by the peer-reviewers****Reorganize the tables for better presentation as suggested by the peer reviewers…****Provide more information in the Conclusions regarding suggestions for future directions pertaining to this research study.****Kindly address all other pertinent issues as suggested by the peer- reviewers.**

We look forward to receiving your revised manuscript.

Kind regards,

Sylvester Chidi Chima, M.D., L.L.M, LLD

Academic Editor

PLOS ONE

Journal Requirements:

"This survey was commissioned by the World Health Organization to provide supporting evidence for the development of the WHO Guidelines on Mental Health at Work. The funding was held by the University of Alberta. JN and PAC were paid from these funds to support this research. WHO representatives participated in the design of the survey and supported its dissemination. Analysis was conducted independently by the study team and presented to WHO. A WHO representative reviewed this manuscript for consistency with WHO terminology but did not influence the findings or conclusions."

"JKM, JN, PAC, GM and AG declared that no competing interests exist.

EEM has received funding from Otsuka-Lundbeck for patient educational activities.

RWL has received honoraria for ad hoc speaking or advising/consulting, or received research

funds, from: Abbvie, Asia-Pacific Economic Cooperation, Bausch, BC Leading Edge

Foundation, Brain Canada, Canadian Institutes of Health Research, Canadian Network for Mood and Anxiety Treatments, CAN-BIND Solutions, Carnot, Grand Challenges Canada, Healthy

Minds Canada, Janssen, Lundbeck, Medscape, Michael Smith Foundation for Health Research,

MITACS, Neurotorium, Ontario Brain Institute, Otsuka, Pfizer/Viatris, Shanghai Mental Health

Center, Sunnybrook Health Sciences Centre, Unity Health, Vancouver Coastal Health Research

Institute, and VGH-UBCH Foundation. "

5. Please include a caption for figure 1. 

Reviewers' comments:

Reviewer's Responses to Questions

**Comments to the Author**

1. Is the manuscript technically sound, and do the data support the conclusions?

Reviewer #1: Yes

Reviewer #2: Yes

2. Has the statistical analysis been performed appropriately and rigorously? 

Reviewer #1: I Don't Know

Reviewer #2: Yes

3. Have the authors made all data underlying the findings in their manuscript fully available?

Reviewer #1: Yes

Reviewer #2: Yes

4. Is the manuscript presented in an intelligible fashion and written in standard English?

Reviewer #1: Yes

Reviewer #2: Yes

5. Review Comments to the Author

Reviewer #1: Overall

This is a well written, interesting study, with an international (including multiple languages) perspective. It is a great foundation for further exploration of workplace mental health programs and guidelines.

Major comments:

Your survey study has quantitative questions alongside a few, simple open-ended text responses – use “open-ended questions” or “comments” instead of referring to “qualitative” throughout the paper. You are using a qualitative data collection technique, not qualitative research methods. Thank you for labeling the quotes appropriately.

How many people were approached to complete the survey? Please provide more detail of how were participants recruited. Were incentives used? Were all participants asked the same questions (managers and workers)? It is unclear in the results section who were the respondents to Theme 1 and Theme 2, and then the “Perspectives of Managers”. Please add a paragraph in the methods describing your recruitment and sample.

Recommended additional analyses by gender – as preferences for males and mental health and workplaces are well known to be different from females. Especially in terms of access (table 12).

Seaton et al. (2017). Men’s mental health promotion interventions: a scoping review. American Journal of men’s health, 11(6), 1823-1837.

Seaton et al. (2018). Mental health promotion in male-dominated workplaces: perspectives of male employees and workplace representatives. Psychology of men and masculinity, 20(4), 541-552.

I see that 68.1% indicated current or prior experience with a mental health condition (page 8), is that participant’s self-experience, or caregiving/providing support to someone with a mental health concern/illness? Please clarify. This would also be a useful and interesting sub analysis.

Minor comments

What is the objective or research question? This should be detailed in the introduction.

Clarify the subheading “Current Impact” to “Current impact of work on mental health” or “Impacts of work”; as it is not the impact of mental health. Page 8

Sort data in tables intentionally (i.e., highest/greatest % at the top). You can also consider changing the likert scale tables to stacked bar figures for more impact (example, table 3 and 13).

“Benefits and concerns” is a large section. This might be better organized if separated with the appropriate subheading; or by personal benefits/concerns vs. population/organizational benefits/concerns; or by type of program?. Page 14-17.

Table 14 & 15, since you asked for people to select all, I would put present raw numbers over %s. These tables and their associated narration (page 14/15) needs some editing for clarity. They are challenging to follow and interpret.

Page 19 – Consider moving the “quality and appropriateness of programs and supports” up to Theme 1; and “engagement and collaboration with workers” up to “Organizational Changes”. No need for a separate section of “Additional considerations from Qualitative Responses”; these 2 topics can be embedded elsewhere.

I think the “Perspectives of Managers” section can be further flushed out, with its’ own theme(s), or else integrated within the other subheadings. There are some similar and related ideas (i.e., preferences for access with/without a manager).

The discussion section flows well and summarizes the main findings. You could consider adding some additional details about next steps, or what the WHO plans to do with the findings on a global scale.

Reviewer #2: Overall comments:

This is an interesting article about an international survey among employees and managers about their knowledge, use and attitude about mental health at work programs and initiatives. The results provide important knowledge about barriers and facilitators for better use of activities that can improve mental health at work. Although the article is well structured and written, I have the following comments to further improve the manuscript before publication.

Abstract:

The abstract is lacking more detailed information about the type of survey (online survey in a worldwide convenience sample) and the size of the survey (how many answered? description of the people answering the survey) to give the reader a better idea about what this article will report about. Although it might be difficult with the word count, a few more of this information should be included.

Considering who actually answered (52% indicate that they play a supervisory role at work), I would describe the sample as consisting of employees and supervisors (rather than employers, which refers more to top-management level). In addition, the formulation “workers, including managers” does not make sense, as there is a clear difference between workers and managers/supervisors. Since results for supervisors/managers are reported separately (page 19) the abstract should reflect this (i.e. “The findings of this study seek to reflect the perspectives of workers and of supervisors…”)

Methods:

More information about the origin of the questions is missing. Were all questions developed for this survey? Or were also validated questions and scales used?

Results:

Line 143: I am not quite sure how to understand this result: “To the question, “What type of impact does your work have generally on your mental health?” respondents reported equally that work has both a positive and negative impact (24.5%)” Does it mean that exactly 24,5% mentioned both positive and negative impact of work? Or that 24,5% mentioned positive and 24,5% mentioned negative impact? Please clarify.

Also, I suggest to report the results to this question differently so it is easier to see that actually more participants answered that work had “a somewhat positive (24.5%) impact on mental health” or “a very positive impact on mental health (15.7%)” (40,2% positive) compared to those that answered that work had “a somewhat negative impact on mental health (19.0%)” or “a very negative impact (6.2%)” (25,2% negative).

In the online material, there are some interesting quotes about the positive and negative impact of work on mental health, which I think would be interesting to report a bit more, as it illustrates that both aspects can be experienced at the same time. It is also important to highlight the positive aspects of work, as it shows that certain aspects of work can actually support mental health and that these positive aspects are therefore a potential that should be supported more – in addition to decreasing the negative aspects. This is also in line with the integrated approach to workplace mental health that the authors refer to (Line 604: LaMontagne et al., 2014).

A more general comment to the result section is that the presentation could be more informative. A number of tables are only presented shortly with regard to what questions were asked, while a presentation of the results that are shown in the table is often missing. I understand that not everything that can be seen in a table needs to be commented in the text to avoid repetition, but some comment that highlights the most important results should be added. This refers especially to tables 4, 5, 6,, 8 and 9. I also suggest presenting the result ordered by frequency, so it is easier to see which answers were given most often.

Page 14, line 231 Benefits and Concerns: Please check table 14 (line 678) which is commented in this part of the result section. It seems to be identical with table 15 (line 683). So the results that are reported in the text cannot be seen in the table.

Page 22, from line 430 the authors write, “We also captured manager experiences and perspective with policies and programs to support work mental health through organizational changes. 45.2% of managers have provided a referral or information to employees about accessing or participating in such an intervention, while 28.8% have not and 17.8% indicate that these policies and programs are unavailable in their workplace.” Please explain what is meant here as I feel that it is the managers that are in the position to develop and implement policies and programs and that this is not an intervention that employees can be referred to, but rather that at least policies are regulations or aims that the workplace is trying to live up to.

Discussion

I agree with the authors that the most striking result of this survey is the enormous impact of stigma which seems to be in the way for many initiatives that would be able to improve mental health at work. The fear of disclosing mental health problems might lead to negative consequences is expressed in many different ways throughout the entire survey. This is especially surprising considering that more than half of the participants identify themselves as managers. Stigma therefore is not only a barrier to faster help for mental health problems for regular employees, but also for supervisors.

The results are in line with what we found in a survey conduced among experts from a variety of European countries and Australia. Hogg, B., Moreno-Alcázar, A., Tóth, M. D., Serbanescu, I., Aust, B., Leduc, C., ... & Amann, B. L. (2023). Supporting employees with mental illness and reducing mental illness-related stigma in the workplace: An expert survey. European Archives of Psychiatry and Clinical Neuroscience, 273(3), 739-753.

The good news is that workplace anti-stigma program seem to work well and should therefore be used much more. Just for your information, we recently published a review about effectiveness of interventions to reduce mental health related stigma in the workplace which confirms the positive findings in the review by Hanisch et al. that you refer to (reference 27) (Tóth, M. D., Ihionvien, S., Leduc, C., Aust, B., Amann, B. L., Cresswell-Smith, J., ... & Purebl, G. (2023). Evidence for the effectiveness of interventions to reduce mental health related stigma in the workplace: a systematic review. BMJ open, 13(2), e067126.). These programs also seem to work when delivered online which has the advantage that also smaller workplaces with less resources for face-to-face activities could use them.

Another striking result is that “almost one half of managers’ report that they had not provided information or referrals for mental health programs and supports because they don’t perceive it to be part of their job.” Page 23, line 455). This might only refer to referral which might be considered as a task that should be done by health experts. Nevertheless, at least in Europe workplaces are obligated to conduct risk assessments of the working environment – including the psychosocial work environment – with the aim to identify risks early, reduce them and of course also provide help to people who are in need. Policy, law and guidance for psychosocial issues in the workplace: an EU perspective - OSHwiki | European Agency for Safety and Health at Work (europa.eu). That might be an aspect that could be mentioned.

Also, I would like to make the authors aware of a new review that investigated the effects of workplace mental health screenings (Strudwick, J., Gayed, A., Deady, M., Haffar, S., Mobbs, S., Malik, A., ... & Harvey, S. B. (2023). Workplace mental health screening: a systematic review and meta-analysis. Occupational and Environmental Medicine.https://oem.bmj.com/content/oemed/early/2023/06/15/oemed-2022-108608.full.pdf) adding to the critical discussion of this type of intervention. The review concludes that screening followed by feedback and advice does not improve employee mental health.

Finally, since organizational level interventions are mentioned several times and are also seen as a promising way to improve mental health at work I would also like to make you aware of recent reviews that have shown that organizational level interventions to reduce burnout seem to work well. See for example Pijpker, R., Vaandrager, L., Veen, E. J., & Koelen, M. A. (2020). Combined interventions to reduce burnout complaints and promote return to work: A systematic review of effectiveness and mediators of change. International journal of environmental research and public health, 17(1), 55.

DeChant, P. F., Acs, A., Rhee, K. B., Boulanger, T. S., Snowdon, J. L., Tutty, M. A., ... & Craig, K. J. T. (2019). Effect of organization-directed workplace interventions on physician burnout: a systematic review. Mayo Clinic Proceedings: Innovations, Quality & Outcomes, 3(4), 384-408.

For a more general overview of reviews of organizational level interventions see:

Aust, B., Møller, J. L., Nordentoft, M., Frydendall, K. B., Bengtsen, E., Jensen, A. B., ... & Jaspers, S. Ø. (2023). How effective are organizational-level interventions in improving the psychosocial work environment, health, and retention of workers? A systematic overview of systematic reviews. Scandinavian Journal of Work, Environment & Health.

Minor comments

Page 5, line 70. Delete “by”

Page 10, table 5: correct to “wellbeing of workers”

Page 13, line 209: I think you mean box 1, not figure 1? At least I could not find figure 1

Page 14, line 221: delete “their”

6. PLOS authors have the option to publish the peer review history of their article (what does this mean?). If published, this will include your full peer review and any attached files.

Reviewer #1: **Yes: **Allison Soprovich

Reviewer #2: No

---

## [Author Response · Author response to Decision Letter 0]

1 Aug 2023

We would like to thank the reviewers for their comprehensive and thoughtful comments on our manuscript. We have provided a comprehensive response to the reviewers' comments in the attached letter, and also provide them below (see attached letter for formatting with responses to reviewers in bold):

Reviewer #1:

 Overall

This is a well written, interesting study, with an international (including multiple languages) perspective. It is a great foundation for further exploration of workplace mental health programs and guidelines.

Thank you for your encouraging comments and recommendations. We have responded to them individually below and have indicated where we have made corresponding revisions in the manuscript. 

Major comments

Your survey study has quantitative questions alongside a few, simple open-ended text responses – use “open-ended questions” or “comments” instead of referring to “qualitative” throughout the paper. You are using a qualitative data collection technique, not qualitative research methods. Thank you for labeling the quotes appropriately.

Thank you for your comment. We have clarified this in the paper by using the term “open-ended questions” as recommended. These edits have been made throughout the paper and in the abstract, with all changes highlighted in yellow. However, the methodological rationale for including these open ended questions was to generate qualitative data to provide a more comprehensive understanding of the perspectives of respondents. We used qualitative thematic analysis to identify themes throughout the data set and describe the analysis and interpretation of results accordingly in the manuscript. 

How many people were approached to complete the survey? Please provide more detail of how were participants recruited. Were incentives used? Were all participants asked the same questions (managers and workers)? It is unclear in the results section who were the respondents to Theme 1 and Theme 2, and then the “Perspectives of Managers”. Please add a paragraph in the methods describing your recruitment and sample.

Thank you for your question. We have clarified on Page 7 and in the abstract that we used convenience sampling to recruit survey participants. We also clarify that we did not offer incentives: 

We used convenience sampling to generate survey participation. This consisted of broad dissemination via several channels, including through international global mental health networks, university networks, via mental health, workplace, and patient-serving organizations, and via social media. No incentives were provided to encourage survey participation.

We have also clarified the data collection process for managers and other sub-groups of interest (Page 5) in the following text:

When participants indicated that they were managers (indicating that they supervise employees), ‘high-risk’ workers (indicating that they work in a health, emergency or humanitarian services), and/or provide mental health support at work, the survey logic directed them to a specific set of questions designed for these participant subgroups, which they responded to in addition to the questions that were available for all survey participants.

Recommended additional analyses by gender – as preferences for males and mental health and workplaces are well known to be different from females. Especially in terms of access (table 12).

Seaton et al. (2017). Men’s mental health promotion interventions: a scoping review. American Journal of men’s health, 11(6), 1823-1837.

Seaton et al. (2018). Mental health promotion in male-dominated workplaces: perspectives of male employees and workplace representatives. Psychology of men and masculinity, 20(4), 541-552.

Thank you for making this recommendation. We certainly agree that disaggregating the results by gender (and indeed other demographic features) would provide important insights into differences in values and preferences related to mental health programs and supports at work by specific sub-populations. However, we believe the presentation of additional results would add to the already substantial length of this manuscript, the purpose of which is to provide an overview of global perspectives related to the values and preferences of workers related to accessing mental health support at work. While additional analysis would certainly be informative, it is outside the scope of this paper. We are exploring further publications of the results and will certainly consider expanding the analysis to account for demographic differences in the future. 

I see that 68.1% indicated current or prior experience with a mental health condition (page 8), is that participant’s self-experience, or caregiving/providing support to someone with a mental health concern/illness? Please clarify. This would also be a useful and interesting sub analysis.

Thank you for your question. This refers to personally experiencing a mental health condition. We have clarified this on Page 9 as follows:

A majority of respondents (68.1%) indicate that they currently or have previously experienced a mental health condition…

Minor comments

What is the objective or research question? This should be detailed in the introduction.

Thank you for raising this concern. We have provided a more detailed description of the objective and research questions in the introduction on Page 4, as follows:

The purpose of this global survey was to capture the perspectives of workers, including managers, in order to understand their values and preferences related to mental health at work. Specifically, we sought to 1) understand workers’ perspectives and preferences related to different types of workplace programs and interventions to promote mental wellbeing and prevent and treat mental health conditions, including their help-seeking preferences, and 2) to assess workers’ awareness of and access to specific types of mental health programs and supports at work.

Clarify the subheading “Current Impact” to “Current impact of work on mental health” or “Impacts of work”; as it is not the impact of mental health. Page 8

Thank you for this suggestion. We have revised the subheading to “Current impact of work on mental health” as suggested (Page 9). 

Sort data in tables intentionally (i.e., highest/greatest % at the top). You can also consider changing the likert scale tables to stacked bar figures for more impact (example, table 3 and 13).

We have revised the tables so that the results are ordered in descending order from lowest to highest. 

“Benefits and concerns” is a large section. This might be better organized if separated with the appropriate subheading; or by personal benefits/concerns vs. population/organizational benefits/concerns; or by type of program?. Page 15-17.

Thank you for your suggestion. We have included sub-headings throughout this section (Pages 17-21). 

Table 14 & 15, since you asked for people to select all, I would put present raw numbers over %s. These tables and their associated narration (page 14/15) needs some editing for clarity. They are challenging to follow and interpret.

Thank you for your suggestion. We have presented these tables using the same format as is used in the WHO’s Evidence Summaries and prefer to maintain this consistency. In the column headings, we have indicated the number of respondents and the total number of responses to reflect the use of “select all” for these survey questions. We do not feel that adding raw numbers instead of percentages would change the interpretation of results by readers. 

Regarding the narration, we have added additional text, as follows. For perceived benefits, we have added (Page 17): 

We asked respondents to rate perceived benefits by intervention type according to the following criteria: convenience, being helpful in promoting or supporting mental health, removing barriers like stigma, affordability for the worker, affordability for the company, and benefiting the morale of the organization.

For concerns, we have added the following text (Page 18):

We also asked respondents to rate their concerns by intervention type according to the following criteria: lack of privacy and confidentiality, fear of judgement or stigma, inconvenience, difficulty in accessing it, being expensive for workers, and being expensive for the company.

Page 19 – Consider moving the “quality and appropriateness of programs and supports” up to Theme 1; and “engagement and collaboration with workers” up to “Organizational Changes”. No need for a separate section of “Additional considerations from Qualitative Responses”; these 2 topics can be embedded elsewhere.

Thank you for this suggestion, we agree that it is better to integrate these sections elsewhere. We have moved both sections to the “Concerns with supporting mental health at work” section (Page 19). 

I think the “Perspectives of Managers” section can be further flushed out, with its’ own theme(s), or else integrated within the other subheadings. There are some similar and related ideas (i.e., preferences for access with/without a manager).

Thank you for this suggestion. We do however believe that this section is appropriate as it is because it provides an overview of the responses to specific survey questions that were directed towards managers related to themes including experiences with and perceived capacity to provide mental health support at work and their needs related to capacity building. We do not believe these findings would easily integrate into other sections of the paper. We did not include open-ended questions directed only at managers, making it difficult to identify additional themes specific to managers. 

The discussion section flows well and summarizes the main findings. You could consider adding some additional details about next steps, or what the WHO plans to do with the findings on a global scale.

Thank you for this suggestion. We have included a summary of broad implications of the findings in the Conclusion section and have added additional text (Page 31) related to the WHO Guidelines, as follows: 

The findings directly informed the development of the WHO Guidelines on Mental Health at Work (18) which were launched in the fall of 2022 and will help to advance workplace mental health support globally.

Because these findings directly informed the development of the Guidelines, making detailed recommendations would duplicate the work of the WHO Guideline Development Group and the content of the Guidelines themselves. We do, however, believe that this manuscript and its findings provide important and more detailed insight into the perspectives of workers that complements the content of the WHO Guidelines. 

Reviewer #2: 

Overall comments

This is an interesting article about an international survey among employees and managers about their knowledge, use and attitude about mental health at work programs and initiatives. The results provide important knowledge about barriers and facilitators for better use of activities that can improve mental health at work. Although the article is well structured and written, I have the following comments to further improve the manuscript before publication.

Thank you for your kind words about this paper. We appreciate your additional comments and have responded to them below, indicating where we have made corresponding changes in the manuscript. 

Abstract:

The abstract is lacking more detailed information about the type of survey (online survey in a worldwide convenience sample) and the size of the survey (how many answered? description of the people answering the survey) to give the reader a better idea about what this article will report about. Although it might be difficult with the word count, a few more of this information should be included.

Thank you for this suggestion. We have included additional details in the abstract within the limitations of the word limit. 

Considering who actually answered (52% indicate that they play a supervisory role at work), I would describe the sample as consisting of employees and supervisors (rather than employers, which refers more to top-management level). In addition, the formulation “workers, including managers” does not make sense, as there is a clear difference between workers and managers/supervisors. Since results for supervisors/managers are reported separately (page 19) the abstract should reflect this (i.e. “The findings of this study seek to reflect the perspectives of workers and of supervisors…”)

Thank you for raising this point of clarification. We have amended the language to state “workers and managers” as we agree it is more accurate. We have made this change in the abstract and also in the Discussion on pages 25. 

Methods:

More information about the origin of the questions is missing. Were all questions developed for this survey? Or were also validated questions and scales used?

Thank you for raising this issue. We have included more detail about the survey development on Page 4-5 as follows: 

The survey development was led by co-first authors JN and JKM, in collaboration with co-author GM and the WHO technical officer responsible for the guideline development process. The survey scope and questions, including the intervention types that were included, were informed by the key questions identified for the Guideline Development Group and an Evidence to Decision Making Framework provided by the WHO partners. Survey questions were developed specifically for this survey based on a review of the relevant literature, and elements of the survey structure was informed by similar surveys including the WHO Consolidated Guideline on Self-care interventions for Health: Sexual and Reproductive Health and Rights Global Values and Preferences Survey (19). In the Results section, we specify where survey items were informed by existing frameworks or classification systems.

Results:

Line 143: I am not quite sure how to understand this result: “To the question, “What type of impact does your work have generally on your mental health?” respondents reported equally that work has both a positive and negative impact (24.5%)” Does it mean that exactly 24,5% mentioned both positive and negative impact of work? Or that 24,5% mentioned positive and 24,5% mentioned negative impact? Please clarify.

Also, I suggest to report the results to this question differently so it is easier to see that actually more participants answered that work had “a somewhat positive (24.5%) impact on mental health” or “a very positive impact on mental health (15.7%)” (40,2% positive) compared to those that answered that work had “a somewhat negative impact on mental health (19.0%)” or “a very negative impact (6.2%)” (25,2% negative).

Thank you for this suggestion. We have revised this paragraph as follows (Page 9):

To the question, “What type of impact does your work have generally on your mental health?” a majority (40.2%) of respondents reported that work a positive impact on their mental health, with 24.5% reporting a somewhat positive impact and 15.7% reporting a very positive impact. 19.0% reported that work has a somewhat negative impact on their mental health, while 6.2% reported a very negative impact. 9.8% responded neutrally, and 0.7% were unsure. 

In the online material, there are some interesting quotes about the positive and negative impact of work on mental health, which I think would be interesting to report a bit more, as it illustrates that both aspects can be experienced at the same time. It is also important to highlight the positive aspects of work, as it shows that certain aspects of work can actually support mental health and that these positive aspects are therefore a potential that should be supported more – in addition to decreasing the negative aspects. This is also in line with the integrated approach to workplace mental health that the authors refer to (Line 604: LaMontagne et al., 2014).

Thank you for suggesting this addition. We had excluded the open-response results in favour of brevity but agree that these responses provide a more nuanced perspective on the relationship between work and mental health. We have added the following section to Pages 10-11:

We provided an open-response option, inviting participants to provide more context related to the impact of work on their mental health. Respondents provided a spectrum of responses, with some describing the positive impact of work on mental health, as described in the following quotation: 

“My job contributes to positive well-being by providing pay, contact with other people, achievement and sense of importance/contribution to society. (P81, Worker, Man)”

Others report a mixed impact of work on their mental health, with work contributing several positive factors but also contributing to stress: 

“My work provides purpose and structure and social relationships which is important for my mental health. It also provides stress and uncertainty at times. There are positive aspects and negative aspects: it carries stress and burnout but also a sense of fulfillment. (P41, Worker, Woman)” 

Finally, other respondents describe a predominantly negative impact of work on mental health, often related to challenging relationships with management, overly demanding workloads and lack of work-life balance, and insufficient compensation: 

“We are overworked with a lot of projects that we are responsible for, too many meetings, few opportunities for growth, feedback not is not acted upon by leadership/management, there is a lack of transparency about decisions and information, underpaid for value added (P 125, Worker, Woman)”

A more general comment to the result section is that the presentation could be more informative. A number of tables are only presented shortly with regard to what questions were asked, while a presentation of the results that are shown in the table is often missing. I understand that not everything that can be seen in a table needs to be commented in the text to avoid repetition, but some comment that highlights the most important results should be added. This refers especially to tables 4, 5, 6,, 8 and 9. I also suggest presenting the result ordered by frequency, so it is easier to see which answers were given most often.

Thank you for this suggestion. We again attempted to promote brevity in the presentation of results given the length of the paper. We do agree that some interpretation of the results displayed in the tables improves readability and have added text to the sections reporting on Table 4 (Page 11), Table 5 (Page 12), Table 6 (Pages 12-13), Table 8 (Page 14), and Table 9 (Page 15). We have also revised the tables to present results in descending order of frequency. 

Page 14, line 231 Benefits and Concerns: Please check table 14 (line 678) which is commented in this part of the result section. It seems to be identical with table 15 (line 683). So the results that are reported in the text cannot be seen in the table.

We sincerely apologize for this error and thank you for pointing it out. We have now included the correct Table 14. 

Page 22, from line 430 the authors write, “We also captured manager experiences and perspective with policies and programs to support work mental health through organizational changes. 45.2% of managers have provided a referral or information to employees about accessing or participating in such an intervention, while 28.8% have not and 17.8% indicate that these policies and programs are unavailable in their workplace.” Please explain what is meant here as I feel that it is the managers that are in the position to develop and implement policies and programs and that this is not an intervention that employees can be referred to, but rather that at least policies are regulations or aims that the workplace is trying to live up to.

We apologize for the lack of clarity here. In the case of organizational changes, manager experience would consist of providing information to an employee about organizational programs or policies that are designed to promote or support mental health at work. As this question was asked in relation to all other programs and support types, we used the term “referral or information” to indicate that managers may refer to a specific program (e.g. psychosocial interventions”) or provide information about a program or policy at the organizational level. We have clarified this by removing the reference to a referral on Page 25 as follows: 

We also captured manager experiences and perspective with policies and programs to support work mental health through organizational changes. 45.2% of managers have provided information to employees about accessing or participating in such an intervention, while 28.8% have not and 17.8% indicate that these policies and programs are unavailable in their workplace. 

Discussion

I agree with the authors that the most striking result of this survey is the enormous impact of stigma which seems to be in the way for many initiatives that would be able to improve mental health at work. The fear of disclosing mental health problems might lead to negative consequences is expressed in many different ways throughout the entire survey. This is especially surprising considering that more than half of the participants identify themselves as managers. Stigma therefore is not only a barrier to faster help for mental health problems for regular employees, but also for supervisors.

The results are in line with what we found in a survey conduced among experts from a variety of European countries and Australia. Hogg, B., Moreno-Alcázar, A., Tóth, M. D., Serbanescu, I., Aust, B., Leduc, C., ... & Amann, B. L. (2023). Supporting employees with mental illness and reducing mental illness-related stigma in the workplace: An expert survey. European Archives of Psychiatry and Clinical Neuroscience, 273(3), 739-753.

The good news is that workplace anti-stigma program seem to work well and should therefore be used much more. Just for your information, we recently published a review about effectiveness of interventions to reduce mental health related stigma in the workplace which confirms the positive findings in the review by Hanisch et al. that you refer to (reference 27) (Tóth, M. D., Ihionvien, S., Leduc, C., Aust, B., Amann, B. L., Cresswell-Smith, J., ... & Purebl, G. (2023). Evidence for the effectiveness of interventions to reduce mental health related stigma in the workplace: a systematic review. BMJ open, 13(2), e067126.). These programs also seem to work when delivered online which has the advantage that also smaller workplaces with less resources for face-to-face activities could use them.

Thank you for your comments and valuable reflections on the findings of this survey, and for sharing your recent publications with us. We have added them to the references in the section on stigma in the Discussion. All new additions to the reference list are also highlighted in yellow. 

Another striking result is that “almost one half of managers’ report that they had not provided information or referrals for mental health programs and supports because they don’t perceive it to be part of their job.” Page 23, line 455). This might only refer to referral which might be considered as a task that should be done by health experts. Nevertheless, at least in Europe workplaces are obligated to conduct risk assessments of the working environment – including the psychosocial work environment – with the aim to identify risks early, reduce them and of course also provide help to people who are in need. Policy, law and guidance for psychosocial issues in the workplace: an EU perspective - OSHwiki | European Agency for Safety and Health at Work (europa.eu). That might be an aspect that could be mentioned.

Thank you for raising this point. We have included the following to acknowledge the diversity in workplace polices by country and region as follow on Page 26:

While workplace policies regarding assessing and monitoring psychological safety and mental health vary by region and country, these findings suggest there is a need for enhanced training and capacity building for managers, including by supporting them to take a proactive role in identifying and supporting employee mental health. Evidence supports the importance of management training to promote mental health at work (25).

Also, I would like to make the authors aware of a new review that investigated the effects of workplace mental health screenings (Strudwick, J., Gayed, A., Deady, M., Haffar, S., Mobbs, S., Malik, A., ... & Harvey, S. B. (2023). Workplace mental health screening: a systematic review and meta-analysis. Occupational and Environmental Medicine.https://oem.bmj.com/content/oemed/early/2023/06/15/oemed-2022-108608.full.pdf) adding to the critical discussion of this type of intervention. The review concludes that screening followed by feedback and advice does not improve employee mental health.

Thank you for calling our attention to this recent study. We have included the reference in the section on screening with the following updates (Page 28):

A US study testing a population-level, opt-in screening intervention with the option to access follow-up care found that the screening program facilitated care uptake among employees (32). However, a 2023 systematic review and meta-analysis (33) found minimal evidence to support workplace mental health screenings and found that screening programs when implemented alone do not support improved mental health. When implemented on a voluntary basis at the population level with adequate privacy and confidentiality measures and available support for those who need it, screening programs may be beneficial in improving early detection and promoting pathways to care, but should not be relied on alone to improve mental health outcomes. 

Finally, since organizational level interventions are mentioned several times and are also seen as a promising way to improve mental health at work I would also like to make you aware of recent reviews that have shown that organizational level interventions to reduce burnout seem to work well. See for example Pijpker, R., Vaandrager, L., Veen, E. J., & Koelen, M. A. (2020). Combined interventions to reduce burnout complaints and promote return to work: A systematic review of effectiveness and mediators of change. International journal of environmental research and public health, 17(1), 55.

DeChant, P. F., Acs, A., Rhee, K. B., Boulanger, T. S., Snowdon, J. L., Tutty, M. A., ... & Craig, K. J. T. (2019). Effect of organization-directed workplace interventions on physician burnout: a systematic review. Mayo Clinic Proceedings: Innovations, Quality & Outcomes, 3(4), 384-408.

For a more general overview of reviews of organizational level interventions see:

Aust, B., Møller, J. L., Nordentoft, M., Frydendall, K. B., Bengtsen, E., Jensen, A. B., ... & Jaspers, S. Ø. (2023). Scandinavian Journal of Work, Environment & Health.

Thank you for calling our attention to these studies. We have added the following text, referencing these studies, to the section on organizational factors (Page 25):

Several studies have found that organizational interventions can be beneficial for employee mental health, including for reducing burnout (25, 26,27), though more research is needed regarding the implementation of these intervention in diverse contexts (27). 

Minor comments

Page 5, line 70. Delete “by”

We have made this change. 

Page 10, table 5: correct to “wellbeing of workers”

Thank you for picking up this typo. We have now corrected it. 

Page 13, line 209: I think you mean box 1, not figure 1? At least I could not find figure 1

Using “Box 1” was an error and we have revised it to be labelled “Figure 1”

Page 14, line 221: delete “their”

We have made this correction (now on Page 16). 

Thank you again for the opportunity to respond to these comments and suggestions. We look forward to any additional comments or suggestions the editor and reviewers might have.

---

## [Decision Letter · Decision Letter 1]

15 Aug 2023

PONE-D-23-05532R1Values and Preferences Related to Workplace Mental Health Programs and Interventions: An International SurveyPLOS ONE

Dear Dr. Murphy,

Thank you for submitting your revised manuscript to PLOS ONE. After careful consideration, we feel that it may be suitable for acceptance for publication. However, there are still a few minor errors that need correction, to fully meet PLOS ONE’s publication criteria as it currently stands. Therefore, we invite you to submit a revised version of the manuscript that addresses the points raised during the review process. 1. Kindly read through your manuscript carefully to correct all typographical and grammatical errors within the manuscript.2. In your revision kindly address all comments and suggestions as recommended by Reviewer 2.

We look forward to receiving your revised manuscript.

Kind regards,

Sylvester Chidi Chima, M.D., L.L.M, LLD.

Academic Editor

PLOS ONE

Journal Requirements:

Reviewers' comments:

Reviewer's Responses to Questions

**Comments to the Author**

1. If the authors have adequately addressed your comments raised in a previous round of review and you feel that this manuscript is now acceptable for publication, you may indicate that here to bypass the “Comments to the Author” section, enter your conflict of interest statement in the “Confidential to Editor” section, and submit your "Accept" recommendation.

Reviewer #1: All comments have been addressed

Reviewer #2: All comments have been addressed

2. Is the manuscript technically sound, and do the data support the conclusions?

Reviewer #1: Yes

Reviewer #2: Yes

3. Has the statistical analysis been performed appropriately and rigorously? 

Reviewer #1: Yes

Reviewer #2: Yes

4. Have the authors made all data underlying the findings in their manuscript fully available?

Reviewer #1: Yes

Reviewer #2: Yes

5. Is the manuscript presented in an intelligible fashion and written in standard English?

Reviewer #1: Yes

Reviewer #2: Yes

6. Review Comments to the Author

Reviewer #1: Thank you for addressing my comments and suggestions adequately. Best wishes and I look forward to reading the final publication.

Reviewer #2: The authors successfully addressed all comments. However, I have a few last comments that I like the authors to check before publication (no need to see the manuscript again)

(I am referring to the page numbers of the manuscript version where changes were marked)

Page 58, Abstract, second and last sentence in the result part:

In the abstract there is still the formulation “employees, including managers” or “workers, including managers” (and also in a new part on page 60, line 30). I recommend again to change this to “workers and their managers” as workers can not include managers.

Also I noticed that the authors sometime use the term “employee”, but I recommend to consistently use the term “worker” for those who are not managers (unless there is a reason to single out “employees”). When you want to refer to all – workers and managers - you could just write “participants of the study” or “respondents”.

The last sentence of the result section seems to be the conclusion. Please add a headline for conclusion.

Page 61, line 57

I think the word “questions” is missing after “open-ended”

Page 63, line 106

Instead of ”Open-ended responses” it should be “Responses to open-ended questions”. (Check also other parts of the manuscript)

Page 65, line 166

add “has” before “a positive impact”

Page 67, line 204

”indicating” instead of ”indication”

Page 70, line 245

“Respondents” instead of “Responses”

Page 70, line 248

“to access them outside of work but not at their home” instead of “to access them outside of work but outside their home”

7. PLOS authors have the option to publish the peer review history of their article (what does this mean?). If published, this will include your full peer review and any attached files.

Reviewer #1: **Yes: **Allison Soprovich

Reviewer #2: No

---

## [Author Response · Author response to Decision Letter 1]

22 Aug 2023

August 22nd, 2023

Dear Dr. Chidi Chima, 

Thank you for the opportunity to make additional revisions and resubmit our manuscript, entitled “Values and Preferences Related to Workplace Mental Health Programs and Interventions: An International Survey” for consideration by PLoS One. 

Below, we provide responses to Reviewer #2’s comments below the original comments in bold. Revisions in the manuscript text have been highlighted in yellow. 

Journal Requirements: 

We have reviewed the reference list and confirm it is complete and correct. To the best of our knowledge we have not included any retracted articles in our reference list. 

Comments from Reviewer #2: 

The authors successfully addressed all comments. However, I have a few last comments that I like the authors to check before publication (no need to see the manuscript again)

(I am referring to the page numbers of the manuscript version where changes were marked)

Page 58, Abstract, second and last sentence in the result part:

In the abstract there is still the formulation “employees, including managers” or “workers, including managers” (and also in a new part on page 60, line 30). I recommend again to change this to “workers and their managers” as workers can not include managers.

Thank you for this comment. We have now updated the language in the abstract as suggested. 

Also I noticed that the authors sometime use the term “employee”, but I recommend to consistently use the term “worker” for those who are not managers (unless there is a reason to single out “employees”). When you want to refer to all – workers and managers - you could just write “participants of the study” or “respondents”.

We have changed the term “employee” to “worker” throughout, except where the term refers to workers that are directly employed by others to distinguish from the broader category of “worker” which may include self-employed, gig, contract or other types of workers. We have made these changes on Page 2, Page 17, Page 20, Page 23, Page 25, Page 26, Page 27, and Page 28. 

The last sentence of the result section seems to be the conclusion. Please add a headline for conclusion.

We believe the reviewer was referring to the last sentence of the Abstract. We have added a heading for “Conclusion” before the last sentence on Page 2 as recommended. 

Page 61, line 57

I think the word “questions” is missing after “open-ended”

Thank you for catching this omission. We have added the word “questions” to line 57 (Page 5 in the Word document of the manuscript). 

Page 63, line 106

Instead of ”Open-ended responses” it should be “Responses to open-ended questions”. (Check also other parts of the manuscript)

We have made this change on line 106 (Page 7 in the Word document). We have also made this change on Line 123 (Page 8), Line 331 (Page 18), Line 424 (Page 21), and on Line 576 (Page 28).

Page 65, line 166

add “has” before “a positive impact”

We have now made this change on Line 166, Page 9. 

Page 67, line 204

”indicating” instead of ”indication”

We have now made this change on Line 204, Page 11. 

Page 70, line 245

“Respondents” instead of “Responses”

We have made this change on line 245 on Page 14. We also changed “indication” to “indicating” on Line 246 of the same page. 

Page 70, line 248

“to access them outside of work but not at their home” instead of “to access them outside of work but outside their home”

We have made this change on Line 248, Page 14. 

Thank you again for the opportunity to respond to these further suggestions. 

Sincerely, 

Dr. Jill Murphy

Research Associate 

Department of Psychiatry 

University of British Columbia

Canada

---

## [Editor Report · Decision Letter 2]

29 Aug 2023

Values and Preferences Related to Workplace Mental Health Programs and Interventions: An International Survey

PONE-D-23-05532R2

Dear Dr. Murphy,

We’re pleased to inform you that your manuscript has been judged scientifically suitable for publication and will be formally accepted for publication once it meets all outstanding technical requirements.

Kind regards,

Sylvester Chidi Chima, M.D., L.L.M. LLD

Academic Editor

PLOS ONE
---

## [Editor Report · Acceptance letter]

1 Sep 2023

PONE-D-23-05532R2 

Values and Preferences Related to Workplace Mental Health Programs and Interventions: An International Survey 

Dear Dr. Murphy:

I'm pleased to inform you that your manuscript has been deemed suitable for publication in PLOS ONE. Congratulations! Your manuscript is now with our production department. 

Kind regards, 

on behalf of

Professor Sylvester Chidi Chima 

Academic Editor

PLOS ONE